



Hysteretic temperature sensitivity of wetland CH$_4$ fluxes explained by substrate
availability and microbial activity
Kuang-Yu Chang*,
Climate and Ecosystem Sciences Division, Lawrence Berkeley National Laboratory,
Berkeley, California, USA
William J. Riley,
Climate and Ecosystem Sciences Division, Lawrence Berkeley National Laboratory,
Berkeley, California, USA
Patrick M. Crill,
Department of Geological Sciences and Bolin Centre for Climate Research, Stockholm
University, Stockholm, Sweden
Robert F. Grant,
Department of Renewable Resources, University of Alberta, Edmonton, Alberta, Canada
Scott R. Saleska
Department of Ecology and Evolutionary Biology, University of Arizona, Tucson,
Arizona, USA
*Corresponding author: Kuang-Yu Chang, ckychang@lbl.gov
Climate and Ecosystem Sciences Division, Lawrence Berkeley National Laboratory
Berkeley, California, USA
Phone: (510) 495-8141
Key words: Methane cycling; microbial dynamics; climate change; IsoGenie Project;
Stordalen Mire



**Abstract**
Methane ($CH_4$) emissions from wetlands are likely increasing and important in global
climate change assessments. However, contemporary terrestrial biogeochemical model
predictions of $CH_4$ emissions are very uncertain, at least in part due to prescribed
temperature sensitivity of $CH_4$ production and emission. While statistically consistent
apparent $CH_4$ emission temperature dependencies have been inferred from meta-analyses
across microbial to ecosystem scales, year-round ecosystem-scale observations have
contradicted that finding. Using flux observations and mechanistic modeling in two
heavily studied high-latitude research sites (Stordalen, Sweden, and Utqiaġvik, Alaska,
USA), we show here that substrate-mediated hysteretic microbial and abiotic interactions
lead to intra-seasonally varying temperature sensitivity of $CH_4$ production and emission.
We find that seasonally varying substrate availability drives lower and higher modeled
methanogen biomass and activity, and thereby $CH_4$ production, during the earlier and
later periods of the thawed season, respectively. Our findings demonstrate the uncertainty
of inferring $CH_4$ emission or production from temperature alone, and highlight the need
to represent microbial and abiotic interactions in wetland biogeochemical models.



## 1. Introduction


Methane ($CH_4$) is the second most important climate forcing gas with at least a

28-fold higher global warming potential (GWP) than carbon dioxide ($CO_2$) over a 100-
year horizon (Myhre, *et al* 2013). Atmospheric $CH_4$ concentrations have more than
doubled since 1750 (Saunois et al., 2016) and have contributed about 20% of the
additional radiative forcing accumulated in the lower atmosphere (Ciais et al., 2013).
Recent assessments have found that $CH_4$ emissions from wetland and other inland waters
are the largest and most uncertain sources affecting the global $CH_4$ budget (Kirschke et
al., 2013; Poulter et al., 2017; Saunois et al., 2016). Such $CH_4$ emissions account for 25
to 32% of current global total $CH_4$ emissions (Saunois et al., 2016) and contribute
substantially to the renewed and sustained atmospheric $CH_4$ growth after 2006 (Saunois
et al., 2017). Increasing $CH_4$ emissions could offset mitigation efforts and accelerate
climate change (Bastviken et al., 2011; Kirschke et al., 2013) due to their strong influence
on the global radiative energy budget (Neubauer and Megonigal, 2015). However, $CH_4$
emission estimates are poorly constrained due to insufficient quality-controlled
measurements (Bastviken et al., 2011; Kirschke et al., 2013; Saunois et al., 2016) and
uncertain model structures and parameterizations (Melton et al., 2013; Wania et al., 2013;
Xu et al., 2016). In fact, simulations in the ongoing Coupled Model Intercomparison
Project Phase 6 (CMIP6; (Eyring et al., 2016)) do not even request wetland $CH_4$ emission
predictions for the historical or 21[st] century periods. A number of knowledge gaps (Xu et
al., 2016) need to be addressed to improve $CH_4$ model representations and thereby $CH_4$
climate feedback predictions (Dean et al., 2018). Such efforts are imperative because,
among other reasons, permafrost degradation resulting from observed global-scale





permafrost warming (Biskaborn et al., 2019) can stimulate organic matter decomposition
(Schuur et al., 2015) that could augment global warming with a strong contribution from
$CH_4$ (Knoblauch et al., 2018).

Many contemporary terrestrial biogeochemical models parameterize $CH_4$

production (or even $CH_4$ emissions) as a static temperature function of net primary
production or heterotrophic respiration (Melton et al., 2013; Wania et al., 2013; Xu et al.,
2016). Such parameterization is supported by recent meta-analyses that indicate a static
and consistent apparent $CH_4$ production and emission temperature dependence across
microbial to ecosystem scales (Yvon-Durocher et al., 2014). However, measurements
collected across sites with nearly identical wetland climate, hydrology, and plant
community compositions suggest large spatial and temporal variability in the ratio
between ecosystem productivity and $CH_4$ emissions (Hemes et al., 2018). Further,
ecosystem-scale $CH_4$ emissions have hysteretic responses to seasonal changes in gross
primary productivity (GPP), water table depth (WTD), and temperature (Brown et al.,
2014; Goodrich et al., 2015; Rinne et al., 2018; Zona et al., 2016), suggesting that $CH_4$
biogeochemistry may not be accurately represented by static relationships. Consequently,
a mechanistic understanding of factors modulating $CH_4$ production and emission rates is
urgently needed to improve the currently uncertain $CH_4$ biogeochemistry
parameterization.

Here, we investigated the impacts of soil thermal and hydrological history on

$CH_4$ emissions to improve understanding of apparent $CH_4$ emission temperature
dependence and inform $CH_4$ model structure and parameterization. We hypothesized that
a static apparent $CH_4$ emission temperature dependence is not sufficient for modeling



CH$_4$ emissions due to substrate-mediated hysteretic microbial and abiotic interactions
(Tang and Riley, 2014) over seasonal time scales. Specifically, we examined temperature
responses of CH$_4$ emission and production rates measured and modeled in two heavily
studied Arctic field sites (Metcalfe et al., 2018): the Stordalen Mire, Sweden (68.2 °N,
19.0 °E) and Utqiaġvik (formerly Barrow), Alaska (71.3 °N, 156.5 °W). We used a
comprehensive biogeochemistry model (*eocsys*) to investigate the observed intra-seasonal
changes in apparent CH$_4$ emission temperature dependence (e.g., Fig. 1) and evaluate the
uncertainty of ignoring substrate-mediated hysteretic microbial and abiotic interactions.
Although observations of increases in CH$_4$ emissions, spatial heterogeneity, and seasonal
dynamics following permafrost degradation have been discussed (Hodgkins et al., 2014;
McCalley et al., 2014; Olefeldt et al., 2013), an understanding of mechanisms regulating
intra-seasonally varying CH$_4$ emissions and their response to temperature is still lacking.
**2. Method**
**2.1 Study site description**
The Stordalen Mire sites are about 10 km east of the Abisko Scientific Research
Station in the discontinuous permafrost zone of northern Sweden and include intact
permafrost palsa, partly thawed bog, and fen (Hodgkins et al., 2014). The mean annual
air temperature and precipitation at the Stordalen Mire are around 0.6 °C and 336 mm y$^{-1}$,
respectively. The measured CH$_4$ emissions are near zero in the palsa due to its deeper
WTD and shallower Active Layer Depth (ALD) (Bäckstrand et al., 2008b, 2008a, 2010);
we therefore did not include this site in our analysis. The bog is ombrotrophic (pH ~4.2)
with WTD fluctuating from the peat surface to 35 cm below the peat surface (Bäckstrand
et al., 2008b, 2008a; Olefeldt and Roulet, 2012), and is dominated by *Sphagnum* spp.





mosses with a moderate abundance of short sedges such as *Eriophorum vaginatum* and
*Carex bigelowii* (Bäckstrand et al., 2008b, 2008a; Malmer et al., 2005; Olefeldt and
Roulet, 2012). The fen is minerotrophic (pH~5.7), has WTD near or above the peat
surface throughout the growing season, and is dominated by tall sedges such as *E.*
*angustifolium*, *C. rostrata* and *Esquisetum* spp. (Bäckstrand et al., 2008b, 2008a; Olefeldt
and Roulet, 2012). The Stordalen Mire bog and fen both have a peat layer ranging from
0.5 to 1 m (Rydén and Kostov, 1980) and an ALD greater than 0.9 m (Bäckstrand et al.,
2008b).

The Utqiaġvik site is located at the Barrow Experimental Observatory at the

northern tip of Alaska's Arctic coastal plain, which is characterized by polygonal
landforms caused by seasonal freezing and thawing of tundra soil (Hinkel et al., 2005).
These polygonal landforms were categorized into separate features based on moisture
variation determined by surface elevations (Wainwright et al., 2015). We analyzed $CH_4$
emissions modeled in the low-centered polygonal landform that was represented as a
connected combination of trough, rim, and center structures (Grant et al., 2017b). The
mean annual air temperature and precipitation at Utqiaġvik are around -12°C and 106
mm $y^{-1}$, respectively. The ALD varies spatially from approximately 20 to 60 cm, which is
influenced by soil texture, vegetation, soil moisture, and inter-annual variability
(Shiklomanov et al., 2010).
**2.2 Field measurements**

A system of six automated gas-sampling chambers made of transparent Lexan

was installed at the Stordalen Mire in 2001 (three in the bog and three in the fen). Each
chamber covered an area of 0.14 $m^2$ (38 cm × 38 cm) with a height of 25–45 cm



depending on the vegetation and the depth of insertion, and was closed for 5 minutes
every 3 hours. In addition, each chamber is instrumented with thermocouples measuring
air and ground surface temperatures, and WTD is measured manually three to five times
per week from June to October each year (McCalley et al., 2014). The system was
updated with a new chamber design similar to that described in (Bubier et al., 2003) in
2011. The new chambers each cover an area of 0.2 m$^2$ (45 cm × 45 cm), with a height
ranging from 15 to 75 cm depending on habitat vegetation. We analyze time- and
chamber- specific daily mean CH$_4$ emissions and ground temperature (when there are at
least six 3-hourly measurements per day) recorded in the Stordalen Mire during the
thawed seasons to identify the observed apparent CH$_4$ emission temperature dependence.
We cannot infer an apparent CH$_4$ emission temperature dependence that only recognizes
temperature effects at the Utqiaġvik site because continuous landform specific (i.e.,
trough-, rim-, and center- specific) measurements are not available there.
**2.3 Apparent temperature dependence calculation**

We quantify the apparent temperature dependencies of daily CH$_4$ emission and

CH$_4$ production by fitting Boltzmann-Arrhenius functions of the form:
$\ln F_i(T) = \overline{E_{a,\iota}} \cdot \left(\frac{-1}{kT}\right) + \varepsilon_{F_i}$                                      (Eq. 1)
where $F_i(T)$ is the rate of CH$_4$ emission (i = 1) and CH$_4$ production (i = 2) at absolute
temperature T. $\overline{E_{a,\iota}}$ (in eV) and $\varepsilon_{F_i}$ correspond to the fitted apparent activation energy
(slope) and base reaction rate (intercept), respectively. $k$ is the Boltzmann constant
(8.62 × 10$^{-5}$ eV K$^{-1}$).

We defined earlier and later periods as the time before and after the modeled (or

measured) temperature (air or soil) reaching its maximum value in a thawed season,



respectively, to investigate intra-seasonal changes in apparent CH₄ emission or
production temperature dependencies. Thawed seasons were defined as the time period
when modeled vertical mean 0-20 cm soil temperatures (or measured air and ground
surface temperatures) are at least 1 °C to avoid low CH₄ emissions in the $0-1$ °C
temperature window that can alter the base reaction rate of our Boltzmann-Arrhenius
functions. The vertical mean $0-20$ cm soil temperature was chosen for our analysis
because CH₄ production in our study site is concentrated in the top 20 cm of soil (Chang
et al., 2019b). Consistent hysteretic temperature responses were derived with above zero
vertical mean $0-20$ cm soil temperatures (i.e., include the $0-1$ °C temperature
window), e.g., Fig. 2 vs. Supplementary Fig. 1.
**2.4 Model description**

The *ecosys* model is a comprehensive biogeochemistry model that explicitly

represents interactions among biogeophysical (i.e., hydrological and thermal),
biogeochemical (including carbon, nitrogen, and phosphorus), plant and microbial
processes. The above-ground processes are represented in multi-specific multi-layer plant
canopies, and the below-ground processes are represented in multiple soil layers with
multiphase subsurface reactive transport. CH₄ production (i.e., acetoclastic and
hydrogenotrophic methanogenesis), CH₄ oxidation, and CH₄ transport (i.e., diffusion,
aerenchyma, and ebullition) are explicitly represented in *ecosys*. The *ecosys* model
operates at variable time steps (~seconds to 1 hour) determined by convergence criteria,
and it can be applied at patch scale (spatially homogenous one-dimensional; e.g., (Chang
et al., 2019a)) and landscape scale (spatially variable two- or three-dimensional; e.g.,
(Grant et al., 2017b, 2017a)). The *ecosys* model has been extensively examined against



field measurements made in 2002–2007 (Chang et al., 2019a) and 2011–2013 (Chang et
al., 2019b) in our study sites at the Stordalen Mire, and 2013 in our study sites at
Utqiagvik (Grant et al., 2017b, 2017a, 2019). A qualitative summary of the *ecosys* model
is provided in the supplementary material to this article, and detailed descriptions are
available in the supplements of (Grant et al., 2017b, 2017a). The *ecosys* model structure
remains unchanged from that in earlier studies.
**2.5 Experimental design**

The primary purpose of this study is to explore the implications of the observed

$CH_4$ emission hysteresis (Fig. 1) and highlight the need to recognize factors other than
temperature that control ecosystem-scale $CH_4$ emissions. We develop a mechanistic
explanation for such hysteresis by investigating how the modeled environmental drivers
modulate $CH_4$ emission hysteresis. The modeled data used in this study are extracted
from our earlier simulations that can be downloaded from the IsoGenie database
(https://isogenie-db.asc.ohio-state.edu/; (Chang et al., 2019a, 2019b)) and the NGEE-
Arctic database (https://ngee-arctic.ornl.gov/; (Grant et al., 2017b, 2017a)). Our analysis
focuses on modeled data because some factors (e.g., root exudates, substrate availability,
and methanogenic population and activity) modulating $CH_4$ production and emission
rates are not continuously measured at our study sites. Our recently published model
results at the Stordalen Mire and Utqiaġvik sites indicate good comparisons with
observations, including for thaw depth ($R^2$ = 0.75 to 0.90), WTD (mean bias = -4.3 to 4.0
cm), and $CO_2$ ($R^2$ = 0.43 to 0.88) and $CH_4$ ($R^2$ = 0.31 to 0.93) surface fluxes (Chang et al.,
2019a, 2019b; Grant et al., 2017b, 2017a, 2019). For conciseness, we focus discussion in
the remainder of the paper on the Stordalen Mire fen site, since it exhibits strong apparent





hysteresis and the underlying mechanisms leading to hysteretic CH$_4$ emissions are similar
across all study sites.

We note the relevant point that the *ecosys* model itself represents temperature

dependence of soil metabolic activity and gas production through locally simulated soil
temperature profiles with an modified Arrhenius function that includes terms for low- and
high-temperature inactivation (Grant, 2015). Besides temperature effects, the *ecosys*
model also represents substrate controls (through Michaelis-Menten kinetics) on
microbial biomass and activity (e.g., Chang et al., 2019b), which is not explicitly
characterized by inferring an apparent whole system temperature dependence (e.g., Eq.
1). These representations allow the model to simulate overall CH$_4$ emission patterns with
more complex dynamics than represented in the apparent temperature dependence
function alone, making it a suitable tool for investigating the relative importance of
temperature dependence versus other factors.
**3. Results and discussion**
**3.1 Observed patterns of apparent CH$_4$ emission hysteresis**

The CH$_4$ emissions measured in the Stordalen Mire bog and fen exhibit

hysteretic responses to ground surface temperature: i.e., at the same ground surface
temperature, greater CH$_4$ emissions during the later than the earlier periods of the thawed
season (Fig. 1). At both sites, plotting time- and chamber- specific CH$_4$ emission and
ground surface temperature measurements from the beginning to end of the thawed
season results in a counterclockwise hysteresis loop at each site-year (2012 to 2017).
Such hysteretic responses lead to intra-seasonally varying apparent CH$_4$ emission
temperature dependencies, suggesting that recognizing temporal variability is needed to





quantify factors modulating $CH_4$ emissions. For example, three distinct apparent $CH_4$
emission temperature dependencies can be derived from the same chamber sampling at
different periods within the same thawed season (i.e., earlier period, later period, and full
season). Despite the high spatial heterogeneity, the observed patterns of $CH_4$ emission
hysteresis are consistent between chambers within and between the bog and fen habitats.
Our results thus demonstrate that $CH_4$ emissions are generally more sensitive to
temperature changes during the later part of the thawed season, and that $CH_4$ emission
strength and temperature dependence vary substantially among site-years. Consistent
hysteretic responses can be found in $CH_4$ emission and air temperature measurements
(Supplementary Fig. 2) and in measurements collected from 2003 to 2008 with relatively
sparse data records (Supplementary Fig. 3). Ignoring the large spatial and temporal
variability in apparent $CH_4$ emission temperature dependencies may not accurately
represent the underlying dynamics, even though the inferred apparent activation energy
for $CH_4$ emissions is comparable between the habitats (e.g., Supplementary Fig. 4).
**3.2 Modeled patterns of apparent $CH_4$ emission hysteresis**

The $CH_4$ emissions modeled by *ecosys*, extracted from our recently published

results in the Stordalen Mire and the Utqiaġvik sites (Chang et al., 2019b; Grant et al.,
2017b), have hysteretic responses to mean 0–20 cm soil temperature (Fig. 2) and air
temperature (Supplementary Fig. 5). The apparent $CH_4$ emission temperature dependence
inferred from the modeled results varies substantially from the beginning to the end of the
thawed season, suggesting that $CH_4$ emissions may not be accurately represented as a
single function of temperature. For each site-year, $CH_4$ emissions modeled in the later
period are greater than those in the earlier period at the same temperature (e.g., Fig. 2),

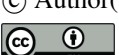



consistent with observations (e.g., Fig. 1). The apparent hysteresis is larger and clearer in
the Stordalen Mire fen compared to the bog and the Utqiaġvik low-centered polygon,
likely from its warmer soil temperatures, shallower WTD, and higher $CH_4$ emissions
(Chang et al., 2019b). In addition to temporal variability, changes in biogeophysical
conditions driven by fine-scale hydrology and vegetation differences can also alter the
apparent functional relationship between $CH_4$ emission and temperature. For example,
apparent $CH_4$ emission temperature dependencies inferred for individual topographic
features (i.e., troughs, rims, and centers) vary substantially within the same wetland
ecosystem at Utqiaġvik (Supplementary Fig. 6), despite being driven by the same
meteorological forcing.

We evaluate the effects of intra-seasonal variability on ecosystem-scale $CH_4$

emissions by estimating apparent $CH_4$ emission temperature dependencies during
different parts of the thawed season. By fitting the Boltzmann-Arrhenius function (Eq. 1)
to the $CH_4$ emissions and 0–20 cm soil temperatures modeled during different time
frames (i.e., earlier period, later period, and full season), we developed and evaluated
three temperature dependence models for each thawed season. Our results show that $CH_4$
emission estimates improve when apparent $CH_4$ emission temperature dependencies were
separately represented in the earlier and later periods, compared to those assuming a
seasonally invariant apparent $CH_4$ emission temperature dependence (Supplementary
Table 1, 2). In the Stordalen Mire, neglecting intra-seasonal variability in apparent $CH_4$
emission temperature dependence leads to overestimated (10 to 81%) and underestimated
(-21 to -40%) $CH_4$ emissions during the earlier and later periods, respectively
(Supplementary Table 1). Consistent prediction bias was found in the Utqiaġvik low-



centered polygon, except in the rims where drier conditions limit CH$_4$ emissions
(Supplementary Table 2).

These results demonstrate that models based on a seasonally invariant apparent

CH$_4$ emission temperature dependence may introduce errors by improperly prescribing
the seasonal dynamics of CH$_4$ biogeochemistry with a static function of temperature. The
substantial intra-seasonal variability, potentially led by site specific thermal and
hydrological history (Updegraff et al., 1998), could be an important and overlooked
property of natural wetlands that currently account for 25 to 32% of global total CH$_4$
emissions (Saunois et al., 2016). Representing intra-seasonally variable apparent CH$_4$
emission or production temperature dependencies in large-scale wetland biogeochemical
models may thus reduce CH$_4$ emission prediction biases and model structural uncertainty.
**3.3 Microbial substrate-mediated CH$_4$ production hysteresis**

For conciseness, we focus our discussion on the potential drivers causing the

hysteretic relationship between CH$_4$ emission and soil temperature modeled in the
Stordalen Mire fen site at 2011, as the underlying mechanisms are consistent across all
site-years. The temporal evolution of CH$_4$ emissions modeled by *ecosys* follows that of
CH$_4$ production (Fig. 3a), with less than 5% of the modeled CH$_4$ production offset by
CH$_4$ oxidation in the Stordalen Mire sites during the thawed season (Chang et al., 2019b).
Modeled CH$_4$ emission (e.g., Fig. 2d) and production (Fig. 3b) rates both exhibit intra-
seasonal variations in their apparent temperature dependencies during the thawed season,
consistent with the varying temperature responses to microbial thermal history reported
in laboratory incubations (Updegraff et al., 1998). The relatively low CH$_4$ oxidation
suggests that hysteretic responses of modeled CH$_4$ emissions to temperature (Fig. 2)





primarily result from hysteretic $CH_4$ production (Fig. 3b) associated with asymmetric
methanogen biomass (Fig. 3c) and activity (Fig. 3d) between the earlier and later periods.
This result is consistent with isotopic measurements which also indicated that changes in
$CH_4$ production, not $CH_4$ oxidation, determine the $CH_4$ emissions observed in the
Stordalen Mire sites (Hodgkins et al., 2014; McCalley et al., 2014).

Increased soil temperatures elevate oxygen demands for aerobic heterotrophs

while reducing oxygen solubility, which favors fermenter and methanogens and thereby
enhance $CH_4$ production. Our model results indicate that the elevated methanogen
biomass and activity during the later period are driven by the increased substrate
availability for methanogenesis later in the thawed season. Modeled substrate
concentrations remain relatively high after peak substrate production rate at maximum
seasonal soil temperature for both acetoclastic (AM; Fig. 4a) and hydrogenotrophic
methanogenesis (HM; Fig. 5a). Relatively high AM (Fig. 4b) and HM (Fig. 5b) substrate
availability during the later period elevates AM and HM energy yields at a given soil
temperature, resulting in higher methanogen growth (Fig. 3d) and biomass (Fig. 3c) later
in the thawed season. Therefore, $CH_4$ production rates during the later period become
higher than those during the earlier period at the same soil temperature (Fig. 3b), which
drives higher $CH_4$ emissions with increased aqueous $CH_4$ concentrations. Although AM
and HM each exhibit microbial substrate-mediated hysteretic temperature responses, AM
appears to be more hysteretic to soil temperature than HM (Fig. 6). The stronger AM
hysteresis is consistent with the larger and clearer $CH_4$ emission hysteresis found in the
Stordalen Mire fen (Fig. 2), where the fractional contribution of AM to total $CH_4$
production is higher than in the Stordalen Mire bog (Chang et al., 2019b; McCalley et al.,



2014). A schematic summarizing the above-mentioned mechanisms for microbial
substrate-mediated $CH_4$ production hysteresis is presented in Fig. 7.
**3.4 Other factors regulating intra-seasonal $CH_4$ emissions**

To evaluate whether our finding that microbial substrate-mediated $CH_4$

production is the primary cause of the observed hysteresis with temperature, we
evaluated four alternative hypotheses: interactions with (1) water table depth; (2) GPP
(via exudation, root litter inputs, and aerenchyma development); (3) thaw depth; and (4)
residual pore-water $CH_4$ concentrations at the end of the earlier part of the thawed season.

First, studies have found that seasonal variations of WTD determine $CH_4$ cycling

dynamics by regulating the temperature response of $CH_4$ emissions, leading to the
observed $CH_4$ emission hysteresis when drought-induced WTD drawdown below the
critical zone for $CH_4$ production (Brown et al., 2014; Goodrich et al., 2015). The
substantial $CH_4$ emission hysteresis observed at the Stordalen Mire fen site is unlikely
caused by seasonal variations in WTD, because the observed WTD are around or above
the peat surface throughout the thawed season with limited effects on $CH_4$ emissions
(Bäckstrand et al., 2008b).

Second, Rinne et al. (2018) reported that the temporal variations of $CH_4$

emissions are strongly regulated by GPP, and the time required to convert GPP to
methanogenesis substrates may cause the observed apparent hysteresis found between
GPP and $CH_4$ emissions. Our results show apparent hysteresis between GPP and $CH_4$
emissions modeled at our study sites (e.g., Fig. 8a), suggesting higher $CH_4$ emissions
later in the thawed season at a given GPP. We next analyzed these interactions using
*ecosys* at the Stordalen Mire fen site to explore whether an apparent hysteretic





relationship between $CH_4$ emissions and GPP is causally connected. We examined three
primary pathways by which GPP could lead to a delayed effect on $CH_4$ emissions, and
thereby hysteresis: increases in (1) fresh carbon inputs from root exudation (Fig. 8b), (2)
below-ground litter inputs (Fig. 8c), and (3) aerenchyma transport caused by GPP-
induced growth of porous sedge roots (Fig. 8d). In contrast to the apparent hysteresis with
GPP, all three of these mechanisms exhibit reversed hysteresis cycles compared to those
between $CH_4$ emissions and temperature. Therefore, these three primary mechanisms are
inconsistent with a causal hysteretic relationship between GPP and $CH_4$ emissions.

Third, studies have suggested that soil temperature increases can expand the

volume of unfrozen soil and thereby stimulate deep carbon decomposition, which can
also contribute to higher carbon emissions later in the thawed season, as has been
observed for upland $CO_2$ emissions (Goulden et al., 1998) and wetland $CH_4$ emissions
(Iwata et al., 2015). Our results show a weak correlation between thaw depth and $CH_4$
emissions during the latter part of the thawed season, although $CH_4$ emissions appear to
increase with deeper thaw during the earlier period (Fig. 8e). Therefore, the hysteretic
relationship between $CH_4$ emission and soil temperature found in our study sites is not
causally connected with the greater volume of unfrozen soil later in the thawed season.
This may be explained by the relatively shallow zone (mostly within the top 20 cm of
soil) of $CH_4$ production (Chang et al., 2019b) compared with the much deeper thaw depth
modeled during the peak $CH_4$ emission period (i.e., July to August) (Chang et al., 2019a).

Fourth, we conducted a sensitivity test, by forcing zero $CH_4$ production during

the later period, to examine the amount of lagged $CH_4$ emissions resulting from $CH_4$
residual stored in the soil profile at the end of the earlier part of the thawed season that



contributes to apparent $CH_4$ emission hysteresis. At the Stordalen Mire fen, later-period
$CH_4$ emissions resulting from earlier-period $CH_4$ residual concentrations decreased
approximately exponentially and contributed about 25% of the $CH_4$ emissions during the
later period (Fig. 9). The timing and magnitude of later-period $CH_4$ emissions attributed
to lagged $CH_4$ emissions do not match with the relatively high $CH_4$ emissions modeled
during the later period. Therefore, our results suggest that lagged $CH_4$ emissions from
residual $CH_4$ produced in the earlier period contribute to, but are not a dominant factor,
modulating the apparent $CH_4$ emission hysteresis.

Collectively, our results suggest that microbial substrate-mediated $CH_4$

production hysteresis is the primary control of the observed apparent $CH_4$ emission
hysteresis. The physical controls on $CH_4$ production and emission (and potentially their
hysteresis patterns) in the sediments of terrestrial freshwater systems may differ from
those we derived from vegetated peat surfaces (Wik et al., 2016), and further
investigation is needed to assess their apparent temperature dependence. To better
understand factors controlling $CH_4$ production and emission, continuous measurements of
seasonal development of methanogenesis substrates and soil temperature at the depth
where $CH_4$ production is prevalent are needed.
**4. Conclusions**

Many contemporary $CH_4$ models parameterize wetland $CH_4$ production (or

emission) as a fixed fraction of net primary productivity or heterotrophic respiration
regulated by a single static function of temperature (Melton et al., 2013; Wania et al.,
2013). Our results suggest that such a parameterization is not accurate because it
oversimplifies microbial responses to changing thermal and hydrological conditions that

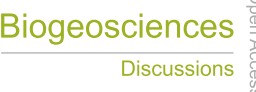

modulate wetland $CH_4$ production and emission rates. More continuous observations
across sites are required to assess model prediction uncertainty and the broader extent to
which our mechanistic explanations apply. In summary, we found that apparent $CH_4$
emission temperature dependencies vary from the earlier to later part of the thawed
season due to substrate-mediated $CH_4$ production hysteresis caused by intra-seasonal
changes in methanogen biomass and activity. We examined four alternative mechanisms
that may contribute to the observed $CH_4$ emission hysteresis with temperature, and found
none of them can exclusively explain the underlying dynamics. Our findings motivate
explicit model representations of microbial dynamics that physiologically link microbial
and abiotic interactions, as only three of 40 recently reviewed $CH_4$ models
mechanistically represent $CH_4$ biogeochemistry (Xu et al., 2016).
**Acknowledgements**
This study was funded by the Genomic Science Program of the United States
Department of Energy Office of Biological and Environmental Research under the
ISOGENIE (DE-SC0016440) and NGEE-Arctic projects under contract DE-AC02-
05CH11231 to Lawrence Berkeley National Laboratory and grants from Swedish VR
(Vetenskaprådet) and Swedish FORMAS to PMC. We acknowledge US National
Science Foundation MacroSystems program (NSF EF 1241037) support for
autochamber measurements between 2013 and 2017. We thank the Abisko Scientific
Research Station of the Swedish Polar Research Secretariat for providing the
meteorological data. The data presented in this study are available at the NGEE Arctic
Database (doi:10.5440/1635534). The *ecosys* source code is available at Zenodo
(doi:10.5281/zenodo.3906642).



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

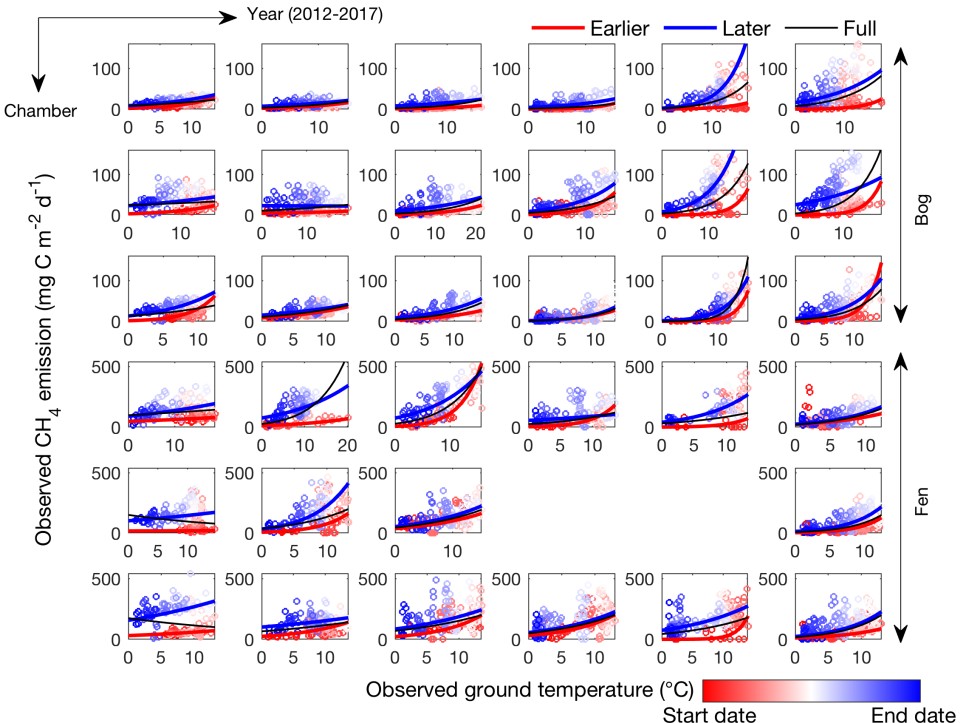


Figure 1. CH$_4$ emissions are hysteretic to ground surface temperature measured in

individual automated chambers in the Stordalen Mire bog (top three panels) and fen

(bottom three panels) sites from 2012 to 2017 thawed seasons (left to right). Open circles

and lines represent the daily data points and the fitted apparent CH$_4$ emission temperature

dependence, respectively. The earlier, later, and full-season periods are colored in red,

blue, and black, respectively. Earlier and later periods are defined as the time before and

after the seasonal maximum ground surface temperature. Start date and end dates

represent the beginning and ending of a thawed season defined as the period when daily

ground surface temperature is above 1 ˚C, respectively.

650

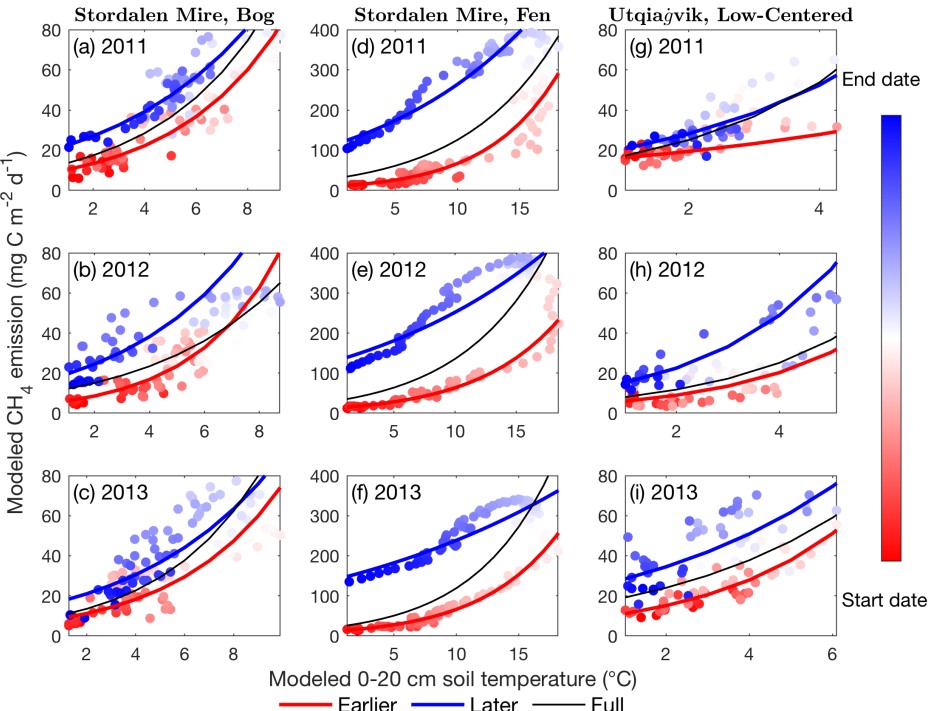

Figure 2. CH₄ emissions are hysteretic to soil temperature modeled in the Stordalen Mire

bog (a to c) and fen (d to f) and the Utqiaġvik low-centered polygon (g to i) from 2011 to

2013 thawed seasons. Dots and lines represent the daily data points and the fitted

apparent temperature dependence, respectively. Earlier, later, and full-season periods are

colored in red, blue, and black, respectively. Earlier and later periods are defined as the

time before and after the seasonal maximum 0-20 cm soil temperature. Start date and end

dates represent the beginning and ending of a thawed season defined as the period when

daily 0-20 cm soil temperature is above 1 ˚C, respectively.





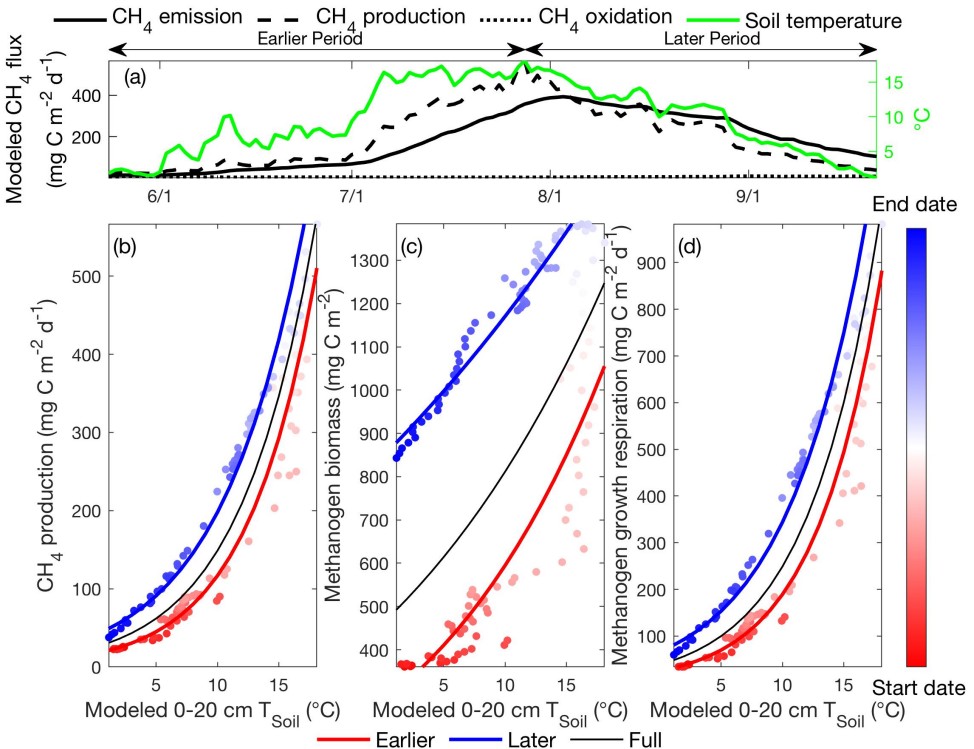

Figure 3. Intra-seasonal variations in apparent $CH_4$ production temperature dependence

result from asymmetric microbial biomass and activity modeled between the earlier and

later periods. Daily $CH_4$ emissions, $CH_4$ production, $CH_4$ oxidation, and 0-20 cm soil

temperature modeled in the Stordalen Mire fen during the 2011 thawed season (a). The

corresponding apparent temperature dependence of the modeled $CH_4$ production (b),

methanogen biomass (c), and methanogen growth respiration (d) during the 2011 thawed

season. Earlier, later, and full-season periods are colored in red, blue, and black,

respectively. Earlier and later periods are defined as the time before and after the seasonal

maximum 0-20 cm soil temperature. Start date and end dates represent the beginning and

ending of a thawed season defined as the period when daily 0-20 cm soil temperature is

above 1 ˚C, respectively.

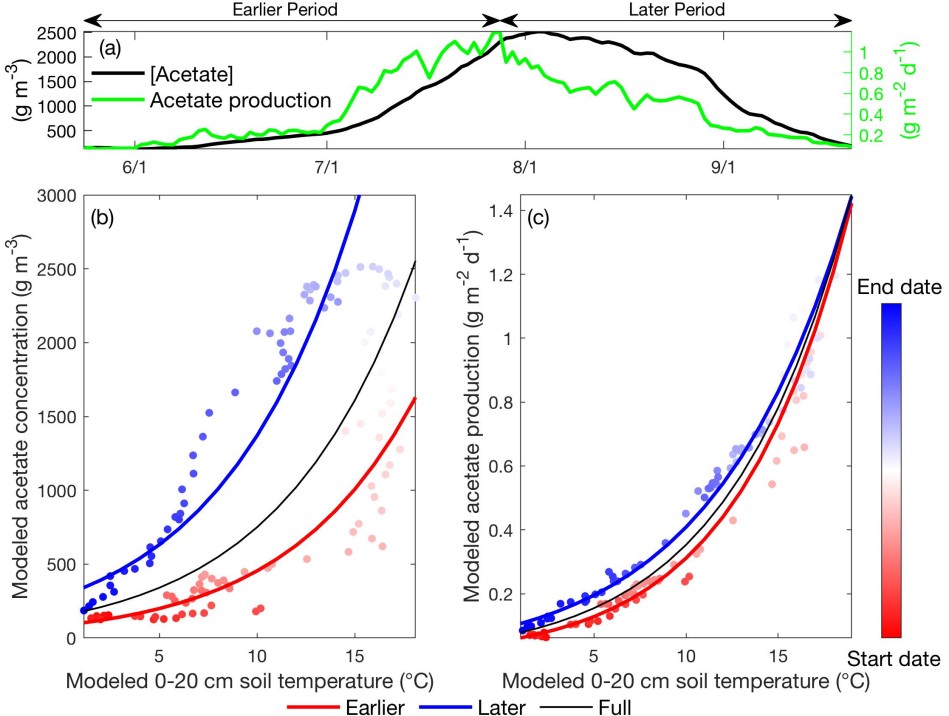

674

Figure 4. Daily acetate concentration and acetate production modeled in the Stordalen

Mire fen during the 2011 thawed season (a). The corresponding apparent temperature

dependence of the modeled acetate concentration (b) and acetate production (c) during

the 2011 thawed season. Dots and lines represent the daily data points and the fitted

apparent temperature dependence, respectively. The earlier, later, and full-season periods

are colored in red, blue, and black, respectively. Earlier and later periods are defined as

the time before and after the seasonal maximum soil temperature (0-20 cm). Start date

and end dates represent the beginning and ending of a thawed season defined as the

period when daily 0-20 cm soil temperature is above 1 ˚C, respectively.

684

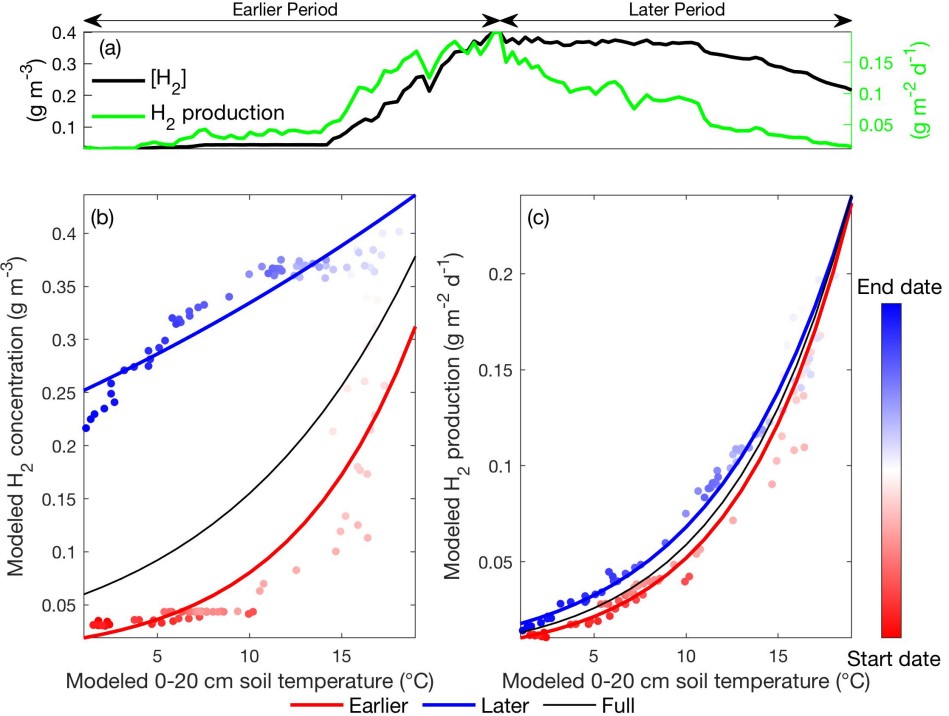

Figure 5. Daily hydrogen concentration and hydrogen production modeled in the

Stordalen Mire fen during the 2011 thawed season (a). The corresponding apparent

temperature dependence of the modeled hydrogen concentration (b) and hydrogen

production (c) during the 2011 thawed season. Dots and lines represent the daily data

points and the fitted apparent temperature dependence, respectively. The earlier, later,

and full-season periods are colored in red, blue, and black, respectively. Earlier and later

periods are defined as the time before and after the seasonal maximum soil temperature

(0-20 cm). Start date and end dates represent the beginning and ending of a thawed

season defined as the period when daily 0-20 cm soil temperature is above 1 ˚C,

respectively.



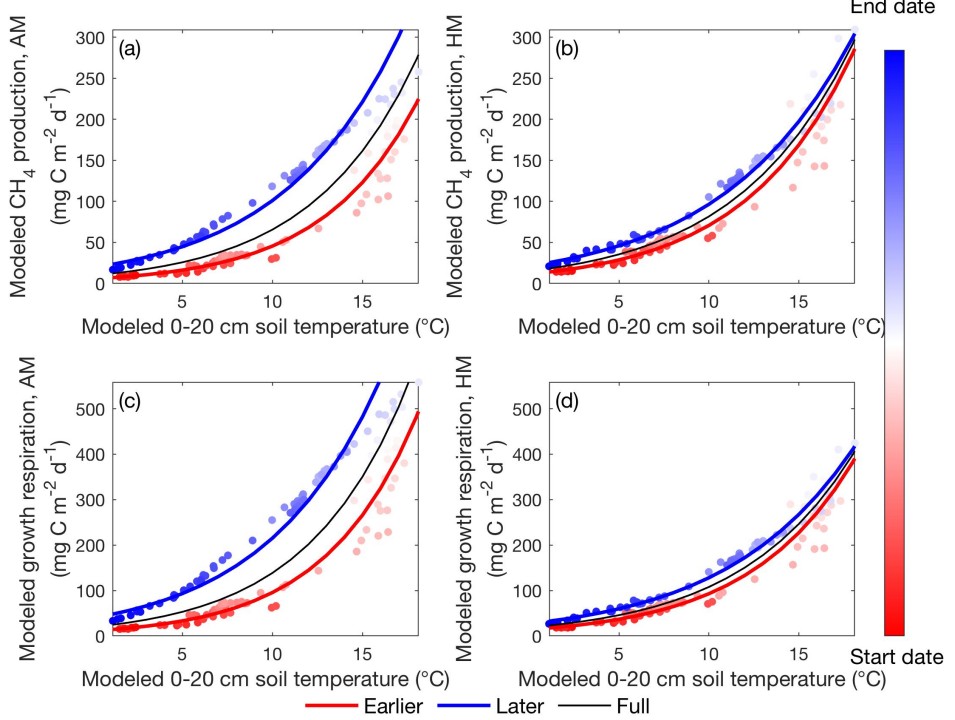

697

Figure 6. Apparent temperature dependence of daily CH$_4$ production for acetoclastic (a)

and hydrogenotrophic (b) methanogenesis, and daily growth respiration for acetoclastic

(c) and hydrogenotrophic (d) methanogens modeled in the Stordalen Mire fen during the

2011 thawed season. Dots and lines represent the daily data points and the fitted apparent

temperature dependence, respectively. The earlier, later, and full-season periods are

colored in red, blue, and black, respectively. Earlier and later periods are defined as the

time before and after the seasonal maximum soil temperature (0-20 cm). Start date and

end dates represent the beginning and ending of a thawed season defined as the period

when daily 0-20 cm soil temperature is above 1 ˚C, respectively.

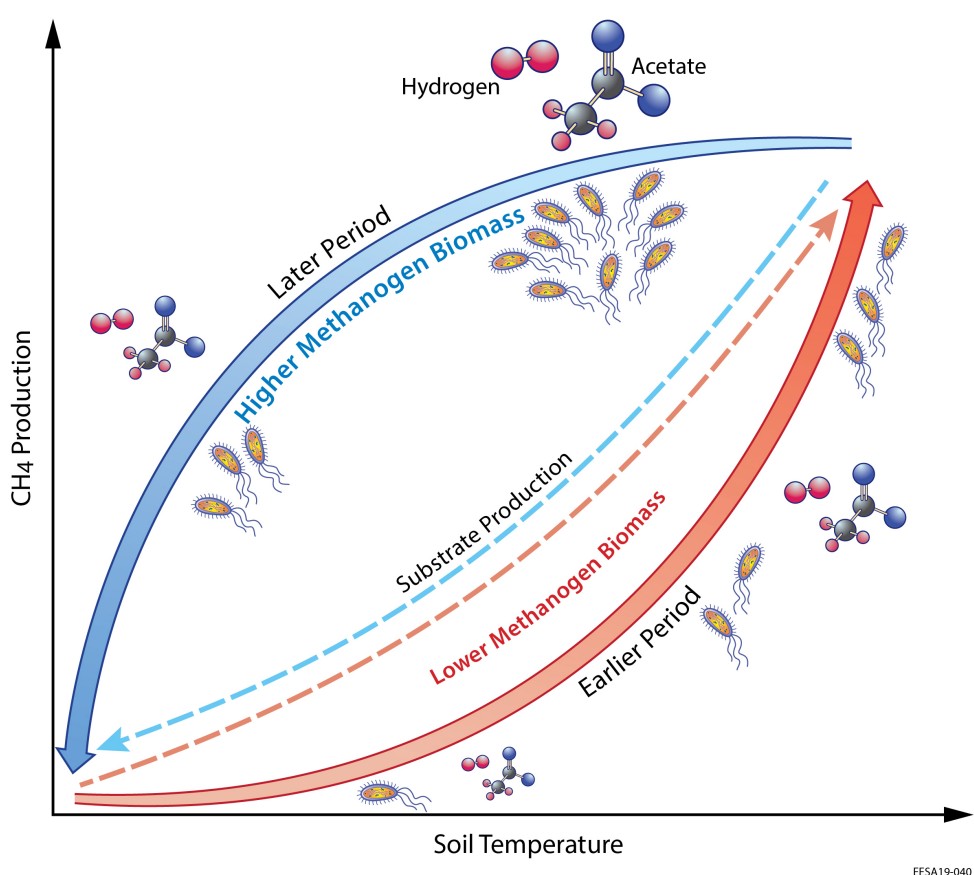

707

Figure 7. Schematic of the microbial substrate-mediated $CH_4$ production hysteresis

proposed in this study. Higher substrate (i.e., acetate and hydrogen) availability

stimulates higher methanogen biomass during the later period, which leads to intra-

seasonal differences in $CH_4$ production between the earlier and later periods.

712





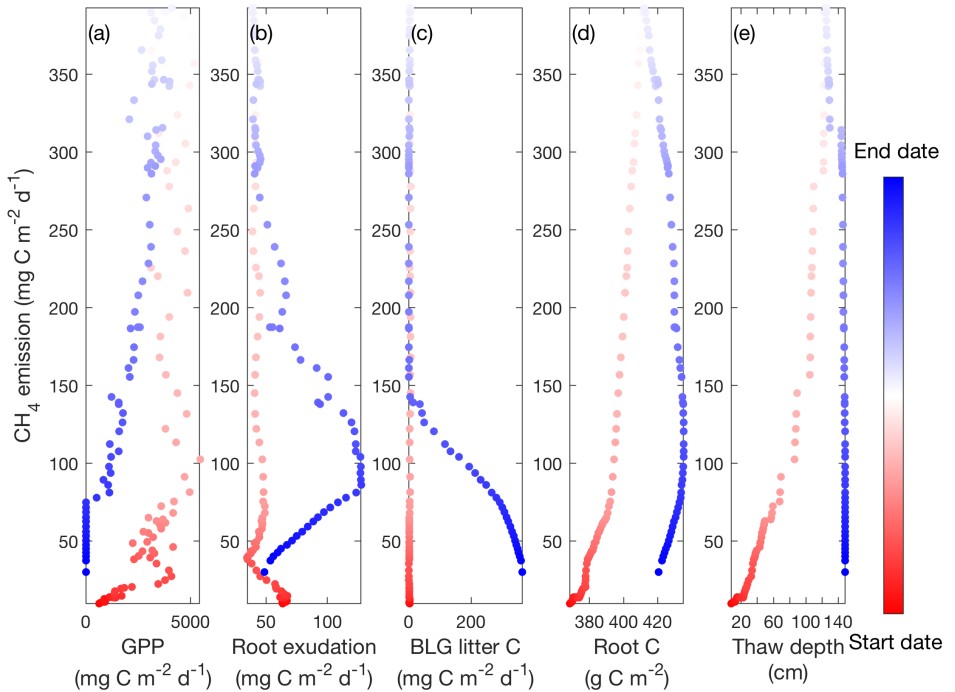

Figure 8. Daily CH₄ emissions have hysteretic responses to gross primary productivity

(a), carbon released from root exudation (b), carbon released from belowground litter

decomposition (c), the amount of root biomass for sedges (d), and thaw depth (e)

modeled in the Stordalen Mire fen during the 2011 thawed season. Dots and lines

represent the daily data points and the fitted apparent temperature dependence,

respectively. Start date and end dates represent the beginning and ending of a thawed

season defined as the period when daily 0-20 cm soil temperature is above 1 °C,

respectively.



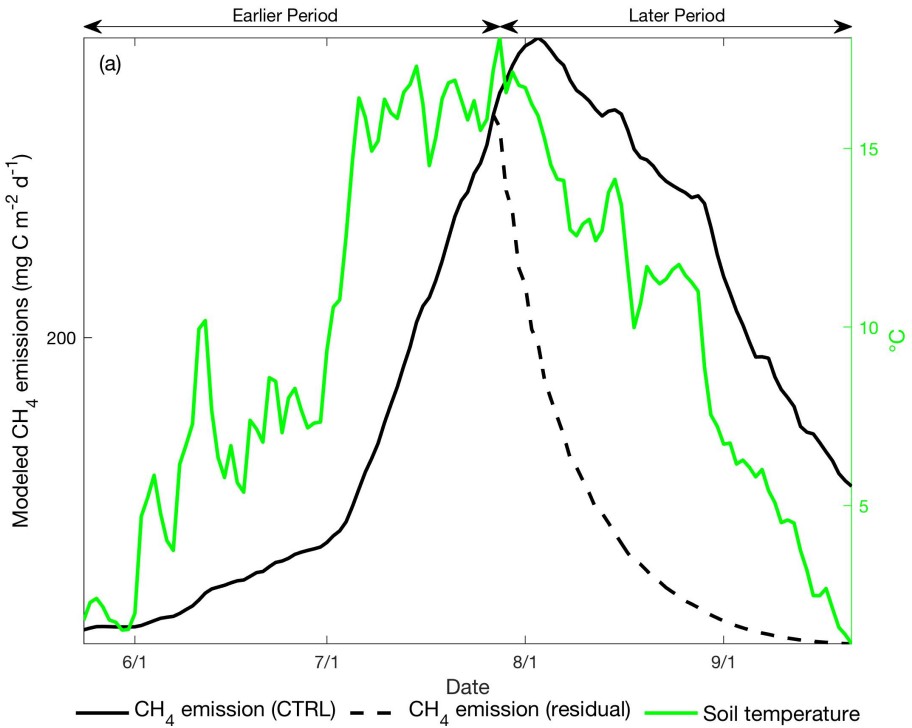

723

Figure 9. Daily CH$_4$ emissions (black line, left axis) and 0-20 cm mean soil temperature

(green line, right axis) modeled in the Stordalen Mire fen during the 2011 thawed season.

Black solid and dashed lines represent the modeled CH$_4$ emissions with and without CH$_4$

production during the later period, respectively. Earlier and later periods are defined as

the time before and after the seasonal maximum soil temperature (0-20 cm).