# Peer review of "Hysteretic temperature sensitivity of wetland CH$_4$ fluxes explained by substrate availability and microbial activity"

_Biogeosciences, 2020_

## Referee Comment (RC1) · Anonymous Referee #1 · 3 Aug 2020

The manuscript 'Hysteretic temperature sensitivity of wetland CH4 fluxes explained by substrate availability and microbial activity' by Chang and co-workers describes a modelling study in which the authors investigate the reasons for the differences in temperature sensitivity of methane emissions at the beginning and the end of the thawing season in two permafrost affected landscapes. They present observational data on this 'hysteretic temperature sensitivity' from one of the investigated sites (Stordalen Mire). However, to investigate the reasons for the observed temperature response they use data generated by their model. Based on the modeling results, the different temperature responses of methane emission during the thawing season is due to higher methanogen biomass and substrate production for methanogenesis in the later

thaw season. This results in higher methane production and emissions at the same temperature in the later season compared to the early seasons.

The manuscript is concerned with a very important topic and there is no doubt that we need a better understanding of the factors regulating the different processes in the wetland CH4 cycle. This improved understanding has to inform models simulating methane emissions and their response to changes in environmental conditions. In this respect, the objective of the current study is highly relevant. On the other side, this study almost exclusively presents model-generated data on the regulation of methane emission in Stordalen Mire. The authors should make this clear and furthermore more critically evaluate the outcome of their model.

First of all, the model simulates a very low contribution of aerobic methane oxidation, which seems to be constant, irrespective of methane production (Fig. 3). The absence of methane oxidation makes the whole story much easier, since in this case, methane emission almost exclusively depend on methane production. However, several studies demonstrated the importance of methane oxidation in Stordalen myre (e.g. Perryman et al., (2020) or Singleton et al., (2018)) and numerous studies on other bogs and fens have shown the utmost importance of methane oxidation for methane emissions. The authors should comment on this, in particular since the unequal importance of methane production and methane oxidation during one thaw season may contribute to the 'hysteretic temperature sensitivity' of methane emissions observed. The bottom soil, where methane production takes place experiences in the early thaw season deeper temperatures than the surface soil, where aerobic processes like methane oxidation take place. In the late season, this pattern is reversed, since the soil starts freezing from the surface, which means that aerobic processes are earlier affected by freezing than anaerobic processes. Therefore, methane oxidizers and methane producers are exposed to different temperatures at the start and the end of the thaw season despite similar mean soil or air temperature. The very low contribution of methane oxidation in their model should be critically discussed on the background of the whole relevant

literature and not only by considering the study supporting their findings. Furthermore, it would be interesting if the model is simulating a substantial contribution of methane oxidation at other sites, e.g. in Barrow.

A second critical point is the simulated extremely high concentration of substrates for methanogens. The simulated maximum acetate concentration is above the substrate concentration that is used to cultivate methanogens in the laboratory. Both simulated acetate and hydrogen concentrations are at least an order of magnitude above those concentrations measured in the presence of active methanogens and also much higher than concentrations that might enable fermenting organisms to gain energy by the production of these end-products. Previous investigations have shown an accumulation of substrates (but not to such high concentrations) if the consumers, in this case the methanogens, are inactive. In case of methanogenic activity, much lower concentrations are present to enable an energy gain for all organisms involved in the anaerobic food chain. Also in this case, the findings should be discussed on the background of available observations.

Furthermore, it is not clearly described in the manuscript, which observations are part of the manuscript. After reading the abstract, I expected observational and mechanistic modelling data from two sites (Strodalen and Barows) but the manuscript indeed presents and discusses almost exclusively model generated data on Stordalen. I suggest more clearly presenting, which kind of observational data are presented. As I understand, only Fig. 1 presents observational data to indicate that the 'hysteretic temperature sensitivity' is real and all the remaining data are generated by the model. I suggest either including more data and discussion on UtqiaÄ ̨avik, or omitting this site. In the current manuscript latter site is only represented in three panels in Fig. 2.

To sum this up: The manuscript lacks in large parts of the discussion a critical evaluation of the model output, which should be discussed on the background of the available observational data.

Specific comments:

L142 -144: The meaning of this sentence is unclear. Please clarify.

L 297: Hodgkins et al. (2014) gives no information on emissions, please revise

L305ff: The energy yield for methanogens indeed increases with rising substrate concentrations but the energy yield of fermenters decreases with rising end-product concentrations. Fermenters will most likely not be able to gain energy from fermentation at such high end-product concentrations. Please consider the whole anaerobic food web.

L329f: In L107 a fluctuating water table between the surface and -35 cm is given. Please clarify.

Cited references:

Hodgkins SB, Tfaily MM, McCalley CK et al. (2014) Changes in peat chemistry associated with permafrost thaw increase greenhouse gas production. Proceedings of the National Academy of Sciences of the United States of America, 111, 5819-5824.

Perryman CR, McCalley CK, Malhotra A et al. (2020) Thaw Transitions and Redox Conditions Drive Methane Oxidation in a Permafrost Peatland. 125, e2019JG005526.

Singleton CM, McCalley CK, Woodcroft BJ et al. (2018) Methanotrophy across a natural permafrost thaw environment. Isme Journal, 12, 2544-2558.

---

## Referee Comment (RC2) · Anonymous Referee #2 · 12 Aug 2020

This study addresses an interesting and important topic in the methane community, the seasonality of CH4 flux, and its causes, emphasizing the thawed period. The study makes use of observational results at two high-latitude sites and previously published modeled results for those sites and further analyzed the differences in CH4 flux and its dependencies on temperature and substrate, microbial biomass before and after the highest temperature. My major comments are as below:

1. The thawed period is used for the analysis; however, it is not clearly defined. I assume it is different from growing season, which is determined based on vegetation. The thaw period is defined with temperature, precisely soil temperature. I did see how it

[Figure]

is defined. As we know that the soil temperature has a very long fluctuation around zero degrees in the shoulder season, how that is used to define the thawed period. Please clarify. 2. The authors used the highest temperature to separate the two periods; this needs to be justified. The strong fluctuation of soil temperature in one year, even the highest degree can be in a few days how to distinguish the temperature difference as < 0.1 degree in two days, particularly when those two similar temperatures are in a few days apart. I think it might be good to use a running average of the soil temperature. 3. Line 154, both air and soil temperature, are used to define the thawed season. It needs a very clear definition on that. In the figure, authors used ground temperature in some places; please keep consistent of air temperature, soil temperature, and ground temperature, which one is used and what it represents. Is the soil temperature < 5cm? is the ground temperature surface temperature? Did air temperature consistent with soil temperature? If not, how are they correlated? How many days of delay in terms of the highest temperature? 4. Although two sites are claimed to be used in the analysis, they are not in equal weight in the analysis. The authors claimed that one site has strong variation, while the other does not. This is not a solid justification. 5. This paper highlights the substrate control, but both acetate and H2 were not validated against to the observational data. How to prove the robustness of the study? Please clarify. 6. As the conceptual diagram shows in figure 7, why the figures 1 – 2 were not plotted in the similar format to clearly show the hysteretic response. The current plotting is not straightforward in terms of supporting the figure 7. 7. Figure 9 might need to be clearly defined, see my previous comments, and put in the first section of the paper. It is the foundation of the whole manuscript. 8. The figure legend of blue color to red color representing the start date to end date, does the highest temperature is in the exact middle of the thawed period? Can you mark the highest temperature on that legend and in the figures? 9. The writing is confusing in some sentences; please revise for clarity purposes. 10. There are a few duplicate references in the bibliography.

---

## Author Comment (AC1) · 5 Sep 2020

The authors would like to thank the Anonymous Reviewer 1 for his/her valuable comments and suggestions to strengthen the analysis presented in our manuscript. The comments and suggestions have been addressed in the revised manuscript (our responses in blue), as follows (reviewer's comments in bold):

The manuscript 'Hysteretic temperature sensitivity of wetland CH4 fluxes explained by substrate availability and microbial activity' by Chang and co-workers describes a modelling study in which the authors investigate the reasons for the differences in temperature sensitivity of methane emissions at the beginning and the end of the thawing season in two permafrost affected landscapes. They present observational data on this 'hysteretic temperature sensitivity' from one of the investigated sites (Stordalen Mire). However, to investigate the reasons for the observed temperature response they use data generated by their model. Based on the modeling results, the different temperature responses of methane emission during the thawing season is due to higher methanogen biomass and substrate production for methanogenesis in the later thaw season. This results in higher methane production and emissions at the same temperature in the later season compared to the early seasons.

The manuscript is concerned with a very important topic and there is no doubt that we need a better understanding of the factors regulating the different processes in the wetland CH4 cycle. This improved understanding has to inform models simulating methane emissions and their response to changes in environmental conditions. In this respect, the objective of the current study is highly relevant. On the other side, this study almost exclusively presents model-generated data on the regulation of methane emission in Stordalen Mire. The authors should make this clear and furthermore more critically evaluate the outcome of their model.

The authors thank the reviewer for the valuable comments. We have carefully revised our manuscript based on the suggestions provided by the reviewer. For example, we have added text in the abstract (Lines 33-36) and introduction (Lines 96-99) indicating that we are using model simulations to interpret the relative role of the complex set of interacting processes responsible for emergent CH4 emissions observed at the Stordalen Mire. We also discuss further observations that would be required to evaluate our proposed mechanisms (Lines 334-342).

First of all, the model simulates a very low contribution of aerobic methane oxidation, which seems to be constant, irrespective of methane production (Fig. 3). The absence of methane oxidation makes the whole story much easier, since in this case, methane emission almost exclusively depend on methane production. However, several studies demonstrated the importance of methane oxidation in Stordalen myre (e.g. Perryman et al., (2020) or Singleton et al., (2018)) and numerous studies on other bogs and fens have shown the utmost importance of methane oxidation for methane emissions. The authors should comment on this, in particular since the unequal importance of methane production and methane oxidation during one thaw season may contribute to the 'hysteretic temperature sensitivity' of methane emissions observed. The bottom soil, where methane production takes place experiences in the early thaw season deeper temperatures than the surface soil, where aerobic processes like methane oxidation take place. In the late season, this pattern is reversed, since the soil starts freezing from the surface, which means that aerobic processes are earlier affected by freezing than anaerobic processes. Therefore, methane oxidizers and methane producers are exposed to different temperatures at the start and the end of the thaw season despite similar mean soil or air temperature. The very low contribution of methane oxidation in their model should be critically discussed on the background of the whole relevant literature and not only by considering the study supporting their findings. Furthermore, it would be interesting if the model is simulating a substantial contribution of methane oxidation at other sites, e.g. in Barrow.

We would like to clarify that the modeled CH4 oxidation rate is not constant, although it appears to be constant in Fig. 3 due to its relatively low values compared with the modeled CH4 production and emission rates. At the Stordalen site, our simulation suggests that CH4 oxidation is unlikely to be a dominant factor controlling thawed-season CH4 emissions (e.g., less than 5% of the modeled CH4 production is oxidized to CO2 from July to August), which is consistent with isotopic evidence from this site (McCalley et al., 2014). Although Perryman et al. (2020) and Singleton et al. (2018) discussed the importance of CH4 oxidation remains uncertain since their results did not estimate the relative strengths of CH4 production and oxidation. For example, Singleton et al. (2018) reported a disconnection between methanotroph abundance and activity in the Stordalen Mire fen, suggesting that the metabolic potential may not necessarily represent microbial activity. In the revised manuscript, we show that seasonal cycles in CH4 production, oxidation, and emission rates modeled at the Stordalen Mire fen are consistent from 2011 to 2013 (SI Fig. 6) while the apparent temperature dependencies of CH4 emissions exhibit

consistent intra-seasonal variation during the corresponding thaw season (Fig. 2d, e, f). Therefore, the hysteretic temperature sensitivity discussed in our study is not caused by the unequal importance of CH4 production and CH4 oxidation modeled in one thaw season. We have included these discussions in the revised manuscript (Lines 298-302; 305-313).

A second critical point is the simulated extremely high concentration of substrates for methanogens. The simulated maximum acetate concentration is above the substrate concentration that is used to cultivate methanogens in the laboratory. Both simulated acetate and hydrogen concentrations are at least an order of magnitude above those concentrations measured in the presence of active methanogens and also much higher than concentrations that might enable fermenting organisms to gain energy by the production of these end-products. Previous investigations have shown an accumulation of substrates (but not to such high concentrations) if the consumers, in this case the methanogens, are inactive. In case of methanogenic activity, much lower concentrations are present to enable an energy gain for all organisms involved in the anaerobic food chain. Also in this case, the findings should be discussed on the background of available observations.

We appreciate the reviewer's comment on the modeled acetate and hydrogen concentrations. We have examined our simulations and found that there was an error in our post processing script converting hourly acetate and hydrogen concentrations modeled at individual soil layers into daily means at the given soil column. We have corrected our post processing script and the corresponding figures (Fig. 4 and 5), and included discussions of available observations and uncertainty of modeled substrate dynamics (Lines 334-342). The acetate concentrations were not within expected ranges, as pointed out by the reviewer.

Furthermore, it is not clearly described in the manuscript, which observations are part of the manuscript. After reading the abstract, I expected observational and mechanistic modelling data from two sites (Strodalen and Barows) but the manuscript indeed presents and discusses almost exclusively model generated data on Stordalen. I suggest more clearly presenting, which kind of observational data are presented. As I understand, only Fig. 1 presents observational data to indicate that the 'hysteretic temperature sensitivity' is real and all the remaining data are generated by the model. I suggest either including more data and discussion on UtqiaÄavik, or omitting this site. In the current manuscript latter site is only

**represented in three panels in Fig. 2.**

**To sum this up: The manuscript lacks in large parts of the discussion a critical evaluation of the model output, which should be discussed on the background of the available observational data.**

We appreciate the reviewer's comment that we need to better indicate which observations are used and how they were used, and to clarify where model simulations are used for analysis. We revised the manuscript according to these reviewer comments (e.g., Lines 30-36), and have included the term 'modeled' at dozens of locations in the manuscript to clarify when modeled results are being discussed. Also, the results collected at Utqiaġvik are now described as a case study to represent the robustness of the modeled CH4 emission hysteresis, where similar hysteretic responses to temperature were found under very different model setup and microclimatic conditions. Although year-round chamber and eddy covariance measurements have indicated hysteretic apparent temperature dependence of CH4 emissions from the beginning to the end of a thaw season, the underlying dynamic remains unclear. Here, we investigated the potential cause of such hysteresis and focused our discussion on modeled results from the Stordalen Mire because we previously validated the modeled CH4 production pathway by the relative abundance of acetoclastic and hydrogenotrophic methanogens inferred from 16S rRNA gene amplicon sequencing data (Chang et al., 2019) (now mentioned on Lines 97-99). Our goal in the manuscript is to propose a CH4 cycling mechanism that has the potential to (1) fill the knowledge gap of the observed  $CH_4$  emission hysteresis, (2) help identify factors should be included in future measurements, and (3) shed light on future CH4 model development.

To address the reviewer's comment regarding a critical evaluation of the model simulations, we acknowledge that additional measurements are needed to further evaluate the cause of the observed CH4 emission hysteresis. However, we note that the substrate mediated CH4 production hysteresis inferred from our model is consistent with the varying temperature responses to microbial thermal history reported in laboratory incubations (Updegraff et al., 1998). We have added text to the revised manuscript to clarify these points (Lines 334-342).

**Specific comments: L142 -144: The meaning of this sentence is unclear. Please clarify.**

Unlike the peatland type specific CH4 emissions measured at the Stordalen Mire, CH4 emissions measured at the Utqiagvik site come from different topographic features with

distinct soil thermal and moisture conditions. To clarify this point, we describe this issue and modify the original Lines 142-144 sentence in the revised manuscript.

**L 297: Hodgkins et al. (2014) gives no information on emissions, please revise**

We have corrected the cited papers used to support this statement (Line 304).

L305ff: The energy yield for methanogens indeed increases with rising substrate concentrations but the energy yield of fermenters decreases with rising end-product concentrations. Fermenters will most likely not be able to gain energy from fermentation at such high end-product concentrations. Please consider the whole anaerobic food web.

We greatly appreciate the reviewer's comment on the modeled acetate and hydrogen concentrations. As described above, we have corrected our post processing script and the modeled acetate and hydrogen concentrations.

**L329f: In L107 a fluctuating water table between the surface and -35 cm is given. Please clarify.**

The water table depth is fluctuating between the peat surface and -35 cm (negative values implying below the peat surface) in the Stordalen Mire bog site (in the original Line 107), and it remains around or above the peat surface in the Stordalen Mire fen site (in the original Lines 111-112 and Lines 329-330). Therefore, we do not expect seasonal variations in water table depth to be the dominant factor controlling CH4 emission hysteresis observed in the Stordalen Mire fen site (Lines 353-356).

**Reference:**

- Chang, K.-Y., Riley, W. J., Brodie, E. L., McCalley, C. K., Crill, P. M., & Grant, R. F. (2019). Methane Production Pathway Regulated Proximally by Substrate Availability and Distally by Temperature in a High-Latitude Mire Complex. *Journal* of Geophysical Research: Biogeosciences, 2019JG005355. https://doi.org/10.1029/2019JG005355
- McCalley, C. K., Woodcroft, B. J., Hodgkins, S. B., Wehr, R. A., Kim, E.-H., Mondav, R., et al. (2014). Methane dynamics regulated by microbial community response to permafrost thaw. *Nature*, 514(7523), 478–481. https://doi.org/10.1038/nature13798
- Perryman, C. R., McCalley, C. K., Malhotra, A., Fahnestock, M. F., Kashi, N. N., Bryce, J. G., et al. (2020). Thaw Transitions and Redox Conditions Drive Methane Oxidation in a Permafrost Peatland. *Journal of Geophysical Research: Biogeosciences*, 125(3). https://doi.org/10.1029/2019JG005526
- Singleton, C. M., McCalley, C. K., Woodcroft, B. J., Boyd, J. A., Evans, P. N., Hodgkins, S. B., et al. (2018). Methanotrophy across a natural permafrost thaw environment. *ISME Journal*, *12*(10), 2544–2558. https://doi.org/10.1038/s41396-018-0065-5
- Updegraff, K., Bridgham, S. D., Pastor, J., & Weishampel, P. (1998). Hysteresis in the

temperature response of carbon dioxide and methane production in peat soils. *Biogeochemistry*, *43*(3), 253–272. https://doi.org/10.1023/A:1006097808262

---

## Author Comment (AC2) · 5 Sep 2020

Please see the attached file

Please also note the supplement to this comment:
https://bg.copernicus.org/preprints/bg-2020-177/bg-2020-177-AC2-supplement.pdf

---

## Author Comment (AC3) · 5 Sep 2020

The authors would like to thank the Anonymous Reviewer 1 for his/her valuable comments and suggestions to strengthen the analysis presented in our manuscript. The comments and suggestions have been addressed in the revised manuscript (our responses in blue), as follows (reviewer's comments in bold):

**The manuscript 'Hysteretic temperature sensitivity of wetland CH4 fluxes explained by substrate availability and microbial activity' by Chang and co-workers describes a modelling study in which the authors investigate the reasons for the differences in temperature sensitivity of methane emissions at the beginning and the end of the thawing season in two permafrost affected landscapes. They present observational data on this 'hysteretic temperature sensitivity' from one of the investigated sites (Stordalen Mire). However, to investigate the reasons for the observed temperature response they use data generated by their model. Based on the modeling results, the different temperature responses of methane emission during the thawing season is due to higher methanogen biomass and substrate production for methanogenesis in the later thaw season. This results in higher methane production and emissions at the same temperature in the later season compared to the early seasons.**

**The manuscript is concerned with a very important topic and there is no doubt that we need a better understanding of the factors regulating the different processes in the wetland CH4 cycle. This improved understanding has to inform models simulating methane emissions and their response to changes in environmental conditions. In this respect, the objective of the current study is highly relevant. On the other side, this study almost exclusively presents model-generated data on the regulation of methane emission in Stordalen Mire. The authors should make this clear and furthermore more critically evaluate the outcome of their model.**

The authors thank the reviewer for the valuable comments. We have carefully revised our manuscript based on the suggestions provided by the reviewer. For example, we have added text in the abstract (Lines 30-36) and introduction (Lines 96-99) indicating that we are using model simulations to interpret the relative role of the complex set of interacting processes responsible for emergent CH$_4$ emissions observed at the Stordalen Mire. We also discuss further observations that would be required to evaluate our proposed mechanisms (Lines 334-342).

We have added the following in the revisions:

Lines 30-36

Here, we show that apparent $CH_4$ emission temperature dependencies inferred from year-round chamber measurements exhibit substantial intra-seasonal variability, suggesting that using static temperature relations to predict $CH_4$ emissions is mechanistically flawed. Our model results indicate that such intra-seasonal variability is driven by substrate-mediated microbial and abiotic interactions: seasonal cycles in substrate availability favors $CH_4$ production later in the season, leading to hysteretic temperature sensitivity of $CH_4$ production and emission.

Lines 96-99

We focus most of the detailed analysis at Stordalen Mire, where we recently validated the modeled $CH_4$ production pathways using acetoclastic and hydrogenotrophic methanogen relative abundance inferred from 16S rRNA gene amplicon sequencing data (Chang et al., 2019).

Lines 334-342

Although the $CH_4$ emission rates and $CH_4$ production pathways modeled in the Stordalen Mire fen have been examined (Chang et al., 2019), continuous substrate concentration measurements are lacking for validating the substrate-mediated hysteretic temperature responses proposed here. Wide ranges of acetate and hydrogen concentrations have been reported from incubation experiments studying methanogenesis (e.g., Hines et al., 2008; Tøsdal et al., 2015; Zhang et al., 2020); however, those values may not be used to validate the time and space specific substrate concentrations modeled at our study sites. Therefore, further studies and additional field measurements are needed to test our proposed hypothesis of the causes of observed $CH_4$ emission hysteresis.

**First of all, the model simulates a very low contribution of aerobic methane oxidation, which seems to be constant, irrespective of methane production (Fig. 3). The absence of methane oxidation makes the whole story much easier, since in this case, methane emission almost exclusively depend on methane production. However, several studies demonstrated the importance of methane oxidation in Stordalen myre (e.g. Perryman et al., (2020) or Singleton et al., (2018)) and numerous studies on other bogs and fens have shown the utmost importance of methane oxidation for methane emissions. The authors should comment on this, in particular since the unequal importance of methane production and methane oxidation during one thaw season may contribute to the 'hysteretic temperature sensitivity' of methane emissions observed. The bottom soil, where methane production takes place experiences in the early thaw season deeper temperatures than the surface soil,**

**where aerobic processes like methane oxidation take place. In the late season, this pattern is reversed, since the soil starts freezing from the surface, which means that aerobic processes are earlier affected by freezing than anaerobic processes. Therefore, methane oxidizers and methane producers are exposed to different temperatures at the start and the end of the thaw season despite similar mean soil or air temperature. The very low contribution of methane oxidation in their model should be critically discussed on the background of the whole relevant literature and not only by considering the study supporting their findings. Furthermore, it would be interesting if the model is simulating a substantial contribution of methane oxidation at other sites, e.g. in Barrow.**

We would like to clarify that the modeled $CH_4$ oxidation rate is not constant, although it appears to be constant in Fig. 3 due to its relatively low values compared with the modeled $CH_4$ production and emission rates. At the Stordalen site, our simulation suggests that $CH_4$ oxidation is unlikely to be a dominant factor controlling thawed-season $CH_4$ emissions (e.g., less than 5% of the modeled $CH_4$ production is oxidized to $CO_2$ from July to August), which is consistent with isotopic evidence from this site (McCalley et al., 2014). Although Perryman et al. (2020) and Singleton et al. (2018) discussed the importance of $CH_4$ oxidation on $CH_4$ cycling, the extent to which $CH_4$ emissions are regulated by $CH_4$ oxidation remains uncertain since their results did not estimate the relative strengths of $CH_4$ production and oxidation. For example, Singleton et al. (2018) reported a disconnection between methanotroph abundance and activity in the Stordalen Mire fen, suggesting that the metabolic potential may not necessarily represent microbial activity. In the revised manuscript, we show that seasonal cycles in $CH_4$ production, oxidation, and emission rates modeled at the Stordalen Mire fen are consistent from 2011 to 2013 (Supplementary Fig. 6) while the apparent temperature dependencies of $CH_4$ emissions exhibit consistent intra-seasonal variation during the corresponding thaw season (Fig. 2d, e, f). Therefore, the hysteretic temperature sensitivity discussed in our study is not caused by the unequal importance of $CH_4$ production and $CH_4$ oxidation modeled in one thaw season. We have included these discussions in the revised manuscript (Lines 298-302; 305-313).

We have added the following in the revisions:

Lines 298-302

Further, the consistent seasonal cycles in $CH_4$ production, oxidation, and emission rates modeled from 2011 to 2013 (Supplementary Fig. 6) indicate that the $CH_4$ emission hysteresis modeled in that period (Fig. 2d, e, f) is not caused by relatively low $CH_4$

oxidation modeled in a particular site-year.

Lines 305-313

Although CH$_4$ oxidation has been proposed to be an important control regulating wetland CH$_4$ emissions, e.g., Perryman et al. (2020) and Singleton et al. (2018), the competitive dynamics between methanogens and methanotrophs throughout the year has not been included in such studies. The modeled CH$_4$ oxidation rate is relatively low during the thawed season when CH$_4$ production is strongest, and relatively high during the shoulder season when CH$_4$ production is weakest (Supplementary Fig. 6). These strong seasonal variations suggest that the relative importance of CH$_4$ production and oxidation on regulating CH$_4$ emissions may fluctuate throughout the year, highlighting the need to properly represent the underlying dynamics controlling CH$_4$ biogeochemistry.

Supplementary Fig. 6

[Figure]

Supplementary Figure 6. Daily $CH_4$ emissions, $CH_4$ production, $CH_4$ oxidation, and $CH_4$ oxidation fraction modeled in the Stordalen Mire fen from 2011 to 2013. $CH_4$ oxidation fraction is defined as the ratio of daily $CH_4$ oxidation to daily $CH_4$ production.

**A second critical point is the simulated extremely high concentration of substrates for methanogens. The simulated maximum acetate concentration is above the substrate concentration that is used to cultivate methanogens in the laboratory. Both simulated acetate and hydrogen concentrations are at least an order of magnitude above those concentrations measured in the presence of active methanogens and also much higher than concentrations that might enable fermenting organisms to gain energy by the production of these end-products. Previous investigations have shown an accumulation of substrates (but not to such high concentrations) if the consumers, in this case the methanogens, are inactive. In case of methanogenic activity, much lower concentrations are present to enable an energy gain for all organisms involved in the anaerobic food chain. Also in this case, the findings should be discussed on the background of available observations.**

We appreciate the reviewer's comment on the modeled acetate and hydrogen concentrations. We have examined our simulations and found that there was an error in our post processing script converting hourly acetate and hydrogen concentrations modeled at individual soil layers into daily means at the given soil column. We have corrected our post processing script and the corresponding figures (Fig. 4 and 5), and included discussions of available observations and uncertainty of modeled substrate dynamics (Lines 334-342). The acetate concentrations were not within expected ranges, as pointed out by the reviewer.

We have added the following in the revisions:

Lines 334-342

Although the $CH_4$ emission rates and $CH_4$ production pathways modeled in the Stordalen Mire fen have been examined (Chang et al., 2019), continuous substrate concentration measurements are lacking for validating the substrate-mediated hysteretic temperature responses proposed here. Wide ranges of acetate and hydrogen concentrations have been reported from incubation experiments studying methanogenesis (e.g., Hines et al., 2008; Tøsdal et al., 2015; Zhang et al., 2020); however, those values may not be used to validate the time and space specific substrate concentrations modeled at our study sites. Therefore, further studies and additional field measurements are needed to test our

proposed hypothesis of the causes of observed CH₄ emission hysteresis.

Fig. 4

[Figure]

Figure 4. Daily acetate concentration and acetate production modeled in the Stordalen Mire fen during the 2011 thawed season (a). The corresponding apparent temperature dependence of the modeled acetate concentration (b) and acetate production (c) during the 2011 thawed season. Dots and lines represent the daily data points and the fitted apparent temperature dependence, respectively. The earlier, later, and full-season periods are colored in red, blue, and black, respectively. Earlier and later periods are defined as the time before and after the seasonal maximum 0-20 cm soil temperature denoted by black cross signs. Start date and end dates represent the beginning and ending of a thawed season defined as the period when modeled daily 0-20 cm soil temperature is above 1 ˚C, respectively.

[Figure]

Figure 5. Daily hydrogen concentration and hydrogen production modeled in the Stordalen Mire fen during the 2011 thawed season (a). The corresponding apparent temperature dependence of the modeled hydrogen concentration (b) and hydrogen production (c) during the 2011 thawed season. Dots and lines represent the daily data points and the fitted apparent temperature dependence, respectively. The earlier, later, and full-season periods are colored in red, blue, and black, respectively. Earlier and later periods are defined as the time before and after the seasonal maximum 0-20 cm soil temperature denoted by black cross signs. Start date and end dates represent the beginning and ending of a thawed season defined as the period when modeled daily 0-20 cm soil temperature is above 1 ˚C, respectively.

**Furthermore, it is not clearly described in the manuscript, which observations are part of the manuscript. After reading the abstract, I expected observational and mechanistic modelling data from two sites (Strodalen and Barows) but the manuscript indeed presents and discusses almost exclusively model generated data**

**on Stordalen. I suggest more clearly presenting, which kind of observational data are presented. As I understand, only Fig. 1 presents observational data to indicate that the 'hysteretic temperature sensitivity' is real and all the remaining data are generated by the model. I suggest either including more data and discussion on UtqiaÄavik, or omitting this site. In the current manuscript latter site is only represented in three panels in Fig. 2.**

**To sum this up: The manuscript lacks in large parts of the discussion a critical evaluation of the model output, which should be discussed on the background of the available observational data.**

We appreciate the reviewer's comment that we need to better indicate which observations are used and how they were used, and to clarify where model simulations are used for analysis. We revised the manuscript according to these reviewer comments (e.g., Lines 30-36), and have included the term 'modeled' at dozens of locations in the manuscript to clarify when modeled results are being discussed. Also, the results collected at Utqiaġvik are now described as a case study to represent the robustness of the modeled $CH_4$ emission hysteresis, where similar hysteretic responses to temperature were found under very different model setup and microclimatic conditions. Although year-round chamber and eddy covariance measurements have indicated hysteretic apparent temperature dependence of $CH_4$ emissions from the beginning to the end of a thaw season, the underlying dynamic remains unclear. Here, we investigated the potential cause of such hysteresis and focused our discussion on modeled results from the Stordalen Mire because we previously validated the modeled $CH_4$ production pathway by the relative abundance of acetoclastic and hydrogenotrophic methanogens inferred from 16S rRNA gene amplicon sequencing data (Chang et al., 2019) (now mentioned on Lines 97-99). Our goal in the manuscript is to propose a $CH_4$ cycling mechanism that has the potential to (1) fill the knowledge gap of the observed $CH_4$ emission hysteresis, (2) help identify factors should be included in future measurements, and (3) shed light on future $CH_4$ model development.

To address the reviewer's comment regarding a critical evaluation of the model simulations, we acknowledge that additional measurements are needed to further evaluate the cause of the observed $CH_4$ emission hysteresis. However, we note that the substrate mediated $CH_4$ production hysteresis inferred from our model is consistent with the varying temperature responses to microbial thermal history reported in laboratory incubations (Updegraff et al., 1998). We have added text to the revised manuscript to clarify these points (Lines 334-342).

**Specific comments:**
**L142 -144: The meaning of this sentence is unclear. Please clarify.**

Unlike the peatland type specific $CH_4$ emissions measured at the Stordalen Mire, $CH_4$ emissions measured at the Utqiaġvik site come from different topographic features with distinct soil thermal and moisture conditions. To clarify this point, we describe this issue and modify the original Lines 142-144 sentence in the revised manuscript.

**L 297: Hodgkins et al. (2014) gives no information on emissions, please revise**

We have corrected the cited papers used to support this statement (Line 304).

This result is consistent with isotopic measurements which also indicated that changes in $CH_4$ production, not $CH_4$ oxidation, determine the $CH_4$ emissions observed in the Stordalen Mire sites (McCalley et al., 2014).

**L305ff: The energy yield for methanogens indeed increases with rising substrate concentrations but the energy yield of fermenters decreases with rising end-product concentrations. Fermenters will most likely not be able to gain energy from fermentation at such high end-product concentrations. Please consider the whole anaerobic food web.**

We greatly appreciate the reviewer's comment on the modeled acetate and hydrogen concentrations. As described above, we have corrected our post processing script and the modeled acetate and hydrogen concentrations.

**L329f: In L107 a fluctuating water table between the surface and -35 cm is given. Please clarify.**

The water table depth is fluctuating between the peat surface and -35 cm (negative values implying below the peat surface) in the Stordalen Mire bog site (in the original Line 107), and it remains around or above the peat surface in the Stordalen Mire fen site (in the original Lines 111-112 and Lines 329-330). Therefore, we do not expect seasonal variations in water table depth to be the dominant factor controlling $CH_4$ emission hysteresis observed in the Stordalen Mire fen site (Lines 353-356).

**Reference:**

Chang, K.-Y., Riley, W. J., Brodie, E. L., McCalley, C. K., Crill, P. M., & Grant, R. F. (2019). Methane Production Pathway Regulated Proximally by Substrate Availability and Distally by Temperature in a High-Latitude Mire Complex. *Journal of Geophysical Research: Biogeosciences*, 2019JG005355. https://doi.org/10.1029/2019JG005355

Hines, M. E., Duddleston, K. N., Rooney-Varga, J. N., Fields, D., & Chanton, J. P. (2008). Uncoupling of acetate degradation from methane formation in Alaskan wetlands: Connections to vegetation distribution. *Global Biogeochemical Cycles*,

*22*(2). https://doi.org/10.1029/2006GB002903

Hodgkins, S. B., Tfaily, M. M., McCalley, C. K., Logan, T. A., Crill, P. M., Saleska, S. R., et al. (2014). Changes in peat chemistry associated with permafrost thaw increase greenhouse gas production. *Proceedings of the National Academy of Sciences*. https://doi.org/10.1073/pnas.1314641111

McCalley, C. K., Woodcroft, B. J., Hodgkins, S. B., Wehr, R. A., Kim, E.-H., Mondav, R., et al. (2014). Methane dynamics regulated by microbial community response to permafrost thaw. *Nature*, *514*(7523), 478–481. https://doi.org/10.1038/nature13798

Perryman, C. R., McCalley, C. K., Malhotra, A., Fahnestock, M. F., Kashi, N. N., Bryce, J. G., et al. (2020). Thaw Transitions and Redox Conditions Drive Methane Oxidation in a Permafrost Peatland. *Journal of Geophysical Research: Biogeosciences*, *125*(3). https://doi.org/10.1029/2019JG005526

Singleton, C. M., McCalley, C. K., Woodcroft, B. J., Boyd, J. A., Evans, P. N., Hodgkins, S. B., et al. (2018). Methanotrophy across a natural permafrost thaw environment. *ISME Journal*, *12*(10), 2544–2558. https://doi.org/10.1038/s41396-018-0065-5

Tøsdal, A., Urich, T., Frenzel, P., & Marianne, M. (2015). Metabolic and trophic interactions modulate methane production by Arctic peat microbiota in response to warming. *Proceedings of the National Academy of Sciences*, E2507–E2516. https://doi.org/10.1073/pnas.1420797112

Updegraff, K., Bridgham, S. D., Pastor, J., & Weishampel, P. (1998). Hysteresis in the temperature response of carbon dioxide and methane production in peat soils. *Biogeochemistry*, *43*(3), 253–272. https://doi.org/10.1023/A:1006097808262

Zhang, L., Liu, X., Duddleston, K., & Hines, M. E. (2020). The Effects of pH, Temperature, and Humic-Like Substances on Anaerobic Carbon Degradation and Methanogenesis in Ombrotrophic and Minerotrophic Alaskan Peatlands. *Aquatic Geochemistry*, (0123456789). https://doi.org/10.1007/s10498-020-09372-0

---

## Author Comment (AC4) · 5 Sep 2020

The authors thank the reviewer for the constructive and helpful comments. We have carefully revised our manuscript based on the suggestions provided by the reviewer, as follows (reviewer's comments in bold):

**This study addresses an interesting and important topic in the methane community, the seasonality of CH4 flux, and its causes, emphasizing the thawed period. The study makes use of observational results at two high-latitude sites and previously published modeled results for those sites and further analyzed the differences in CH4 flux and its dependencies on temperature and substrate, microbial biomass before and after the highest temperature. My major comments are as below:**

**1. The thawed period is used for the analysis; however, it is not clearly defined. I assume it is different from growing season, which is determined based on vegetation. The thaw period is defined with temperature, precisely soil temperature. I did see how it is defined. As we know that the soil temperature has a very long fluctuation around zero degrees in the shoulder season, how that is used to define the thawed period. Please clarify.**

We defined thawed season as the period when the temperature being analyzed is above 1 °C (L 152-155). We agree with the reviewer that the length of thawed season may vary substantially with different temperature thresholds; however, our finding that $CH_4$ production becomes higher later in the thawed season is not sensitive to the definition of thawed season. For example, consistent $CH_4$ emission hysteresis is observed when pairing measured $CH_4$ emissions with soil surface temperature (when soil surface temperature is above 1 °C, Fig. 1) and air temperature (when air temperature is above 1 °C, Supplementary Fig. 1). We have examined the sensitivity of the daily mean 0-20 cm soil temperature used in our thawed season definition, and found consistent hysteretic temperature responses when using 1 °C (Fig. 2) and 0 °C (Supplementary Fig. 3) as the temperature threshold. We have improved the clarity of our thawed season definition to address the reviewer's concern (L 153-159).

We have added the following in the revisions:

L 152-159

Thawed seasons were defined as the time period when measured or modeled temperatures are at least 1 °C to avoid low $CH_4$ emissions in the $0 - 1$ °C temperature window that can alter the base reaction rate of our Boltzmann-Arrhenius functions. Four types of temperature were used in our analysis: (1) measured soil surface temperature (e.g., Fig. 1), (2) modeled vertical mean $0 - 20$ cm soil temperature (e.g., Fig. 2), (3) measured air temperature (e.g., Supplementary Fig. 1), and (4) modeled air temperature (e.g., Supplementary Fig. 2).

Supplementary Fig. 1

[Figure]

Supplementary Figure 1. CH₄ emissions are hysteretic to air temperature measured in individual automated chambers in the Stordalen Mire bog (top three panels) and fen (bottom three panels) sites from 2012 to 2017 thawed seasons (left to right). Open circles and lines represent the daily data points and the fitted apparent CH₄ emission temperature dependence, respectively. The earlier, later, and full-season periods are colored in red, blue, and black, respectively. Earlier and later periods are defined as the time before and after the seasonal maximum air temperature denoted by black cross signs. Start date and end dates represent the beginning and ending of a thawed season defined as the period when measured daily air temperature is above 1 ˚C, respectively.

[Figure]

Supplementary Figure 2. CH$_4$ emissions are hysteretic to air temperature modeled at the Stordalen Mire bog (a to c) and fen (d to f) and the Utqiaġvik low-centered polygon (g to i) from 2011 to 2013 thawed seasons. Dots and lines represent the daily data points and the fitted apparent temperature dependence, respectively. Earlier, later, and full-season periods are colored in red, blue, and black, respectively. Earlier and later periods are defined as the time before and after the seasonal maximum air temperature denoted by black cross signs. Start date and end dates represent the beginning and ending of a thawed season defined as the period when modeled daily air temperature is above 1 ˚C, respectively.

**2. The authors used the highest temperature to separate the two periods; this needs to be justified. The strong fluctuation of soil temperature in one year, even the highest degree can be in a few days how to distinguish the temperature difference as < 0.1 degree in two days, particularly when those two similar temperatures are in a few days apart. I think it might be good to use a running average of the soil temperature.**

The earlier and later periods of a thawed season is separated by the highest daily temperature observed or modeled in that season; however, it is just a qualitative measure describing the intra-seasonal variability detected in apparent temperature dependence of

CH₄ emissions (i.e., quantifying the counterclockwise hysteresis loop shown in the scatters in Fig. 1 and 2). To address this reviewer comment, we have included the temporal variations in apparent temperature dependence of $CH_4$ emissions at weekly timescales (Supplementary Fig. 4) and also found higher $CH_4$ emissions later in a thawed season at the same temperature. Therefore, the hysteretic apparent temperature dependence of $CH_4$ emissions found in our study is not sensitive to the selection of earlier and later periods, nor to the temporal resolution used in representing the process. We have included the discussions above in the revised manuscript (L 253-255) to clarify this point.

We have added the following in the revisions:

L 253-255

Consistent hysteresis patterns are found at weekly timescales (Supplementary Fig. 4), suggesting that the apparent $CH_4$ emission hysteresis is not sensitive to temporal resolution nor the timing of maximum seasonal temperature.

Supplementary Fig. 4

[Figure]

Supplementary Figure 4. Weekly $CH_4$ emissions are hysteretic to weekly soil temperature modeled in the Stordalen Mire bog (a to c) and fen (d to f) and the Utqiaġvik lowcentered polygon (g to i) from 2011 to 2013 thawed seasons. Dots and lines represent the daily data points and the fitted apparent temperature dependence, respectively. Earlier, later, and full-season periods are colored in red, blue, and black, respectively. Earlier and later periods are defined as the time before and after the seasonal maximum 0-20 cm soil temperature denoted by black cross signs. Start date and end dates represent the beginning and ending of a thawed season defined as the period when modeled weekly 0-20 cm soil temperature is above 1 ˚C, respectively.

**3. Line 154, both air and soil temperature, are used to define the thawed season. It needs a very clear definition on that. In the figure, authors used ground temperature in some places; please keep consistent of air temperature, soil temperature, and ground temperature, which one is used and what it represents. Is the soil temperature < 5cm? is the ground temperature surface temperature? Did air temperature consistent with soil temperature? If not, how are they correlated? How many days of delay in terms of the highest temperature?**

We have improved our descriptions in the type of temperature used in our analysis (L 155-159). Our results showed that $CH_4$ emissions are hysteretic to both air and soil temperatures at different temporal resolutions (e.g., Fig. 1 and Supplementary Fig. 1), suggesting that the $CH_4$ emission hysteresis is more sensitive to seasonal cycles in temperature than short-term variations in temperature (e.g., time lags between air and soil temperatures).

**4. Although two sites are claimed to be used in the analysis, they are not in equal weight in the analysis. The authors claimed that one site has strong variation, while the other does not. This is not a solid justification.**

We have revised the manuscript to specify that we are presenting a detailed analysis in the Stordalen Mire fen site, although similar hysteresis patterns can be found in the Stordalen Mire bog and Utqiaġvik sites (L 30-36; 93-99). We address this reviewer comment, which also was pointed out by Reviewer #1, by indicating that results collected at Utqiaġvik are described as a case study to represent the robustness of the modeled $CH_4$ emission hysteresis, because similar hysteretic responses to temperature were found at that site also. An important reason that we focused our discussion on the Stordalen Mire is that we previously validated the modeled $CH_4$ production pathway by the relative abundance of acetoclastic and hydrogenotrophic methanogens inferred from 16S rRNA gene amplicon sequencing data (Chang et al., 2019), which is now mentioned on L 97-99.

We have added the following in the revisions:

L 30-36

Here, we show that apparent $CH_4$ emission temperature dependencies inferred from year-round chamber measurements exhibit substantial intra-seasonal variability, suggesting that using static temperature relations to predict $CH_4$ emissions is mechanistically flawed. Our model results indicate that such intra-seasonal variability is driven by substrate-mediated microbial and abiotic interactions: seasonal cycles in substrate availability favors $CH_4$ production later in the season, leading to hysteretic temperature sensitivity of $CH_4$ production and emission.

L 93-99

We used a comprehensive biogeochemistry model (*ecosys*) to investigate observed intra-seasonal changes in apparent $CH_4$ emission temperature dependence at two high-latitude sites: Stordalen Mire (68.2 °N, 19.0 °E) and Utqiaġvik (formerly Barrow, 71.3 °N, 156.5 °W). We focus most of the detailed analysis at Stordalen Mire, where we recently validated the modeled $CH_4$ production pathways using acetoclastic and hydrogenotrophic methanogen relative abundance inferred from 16S rRNA gene amplicon sequencing data (Chang et al., 2019).

**5. This paper highlights the substrate control, but both acetate and H2 were not validated against to the observational data. How to prove the robustness of the study? Please clarify.**

The temporal changes in $CH_4$ production dynamics and the relative abundance of acetoclastic and hydrogenotrophic methanogens modeled in the Stordalen Mire have been validated in (Chang et al., 2019), suggesting that the model can reasonably represent the observed seasonal cycles in $CH_4$ cycling. Although the substrate mediated $CH_4$ production hysteresis inferred from our model data is consistent with laboratory incubations (Updegraff et al., 1998), we do not have acetate and hydrogen measurements to support the seasonal cycles in modeled substrate concentrations. We have revised the manuscript to clarify that the aim of this model-based study is to shed light on future $CH_4$ model development (i.e., substrate dynamics should be properly represented), and further measurements are required to examine the substrate mediated $CH_4$ production hysteresis proposed here (L334-342).

We have added the following in the revisions:

L334-342

Although the $CH_4$ emission rates and $CH_4$ production pathways modeled in the Stordalen Mire fen have been examined (Chang et al., 2019), continuous substrate concentration measurements are lacking for validating the substrate-mediated hysteretic temperature responses proposed here. Wide ranges of acetate and hydrogen concentrations have been reported from incubation experiments studying methanogenesis (e.g., Hines et al., 2008; Tøsdal et al., 2015; Zhang et al., 2020); however, those values may not be used to

validate the time and space specific substrate concentrations modeled at our study sites. Therefore, further studies and additional field measurements are needed to test our proposed hypothesis of the causes of observed CH₄ emission hysteresis.

**6. As the conceptual diagram shows in figure7, why the figures 1 – 2 were not plotted in the similar format to clearly show the hysteretic response. The current plotting is not straightforward in terms of supporting the figure 7.**

The Arrhenius fits were included in Fig. 1 and 2 to quantify the differences in apparent activation energy for CH₄ emissions inferred from different periods, and to make it easier to compare with previously published data (e.g., Yvon-Durocher et al., (2014)). In the revised manuscript, we use lighter colors for the Arrhenius fits and highlight the counterclockwise apparent hysteresis in the scatters to make it more intuitive to compare with the conceptual diagram Fig. 7

We have added the following in the revisions:

Fig 1

[Figure]

Figure 1. CH₄ emissions are hysteretic to soil surface temperature measured in individual automated chambers at the Stordalen Mire bog (top three panels) and fen (bottom three panels) sites from 2012 to 2017 thawed seasons (left to right). Open circles and lines represent the daily data points and the fitted apparent CH₄ emission temperature dependence, respectively. The earlier, later, and full-season periods are colored in red, blue, and black, respectively. Earlier and later periods are defined as the time before and after the seasonal maximum soil surface temperature denoted by black cross signs. Start date and end dates represent the beginning and ending of a thawed season defined as the period when measured daily soil surface temperature is above 1 ˚C, respectively.

Fig 2

[Figure]

Figure 2. CH₄ emissions are hysteretic to soil temperature modeled in the Stordalen Mire bog (a to c) and fen (d to f) and the Utqiaġvik low-centered polygon (g to i) from 2011 to 2013 thawed seasons. Dots and lines represent the daily data points and the fitted apparent temperature dependence, respectively. Earlier, later, and full-season periods are colored in red, blue, and black, respectively. Earlier and later periods are defined as the time before and after the seasonal maximum 0-20 cm soil temperature denoted by black cross signs. Start date and end dates represent the beginning and ending of a thawed season defined as the period when modeled daily 0-20 cm soil temperature is above 1 ˚C, respectively.

**7. Figure 9 might need to be clearly defined, see my previous comments, and put in the first section of the paper. It is the foundation of the whole manuscript.**

We have improved the description of Fig. 9 based on the reviewer's comments (L 387-389; 394-397). We agree with the reviewer that it is important to point out that the observed $CH_4$ emission hysteresis is unlikely caused by delayed $CH_4$ emissions. Nevertheless, we would like to keep the current structure because it may be more straightforward to people that are not familiar with this research field.

We have added the following in the revisions:

L 387-389

In the sensitivity test, we turned off $CH_4$ production during the later part of the thawed season so the later-period $CH_4$ emissions modeled in this run are driven by lagged releases of earlier-period $CH_4$ production.

L 394-397

Therefore, our results suggest that lagged $CH_4$ emissions from residual $CH_4$ produced in the earlier period are not a dominant factor leading to the observed $CH_4$ emission hysteresis, although lagged $CH_4$ emissions may amplify the apparent $CH_4$ emission hysteresis detected in the system.

**8. The figure legend of blue color to red color representing the start date to end date, does the highest temperature is in the exact middle of the thawed period? Can you mark the highest temperature on that legend and in the figures?**

We have revised the figures so that the highest temperature is in the exact middle of the color bar for the thawed period. The highest temperature has been marked with black crosses in each subplot.

**9. The writing is confusing in some sentences; please revise for clarity purposes.**

We have reviewed the manuscript and improved the writing to address the reviewer's concern on clarity. Please refer to the highlighted texts in the revised manuscript.

**10. There are a few duplicate references in the bibliography.**

We have reviewed the bibliography and fixed the duplicate references.

Reference:

Chang, K.-Y., Riley, W. J., Brodie, E. L., McCalley, C. K., Crill, P. M., & Grant, R. F. (2019). Methane Production Pathway Regulated Proximally by Substrate Availability and Distally by Temperature in a High-Latitude Mire Complex. *Journal*

*of Geophysical Research: Biogeosciences*, 2019JG005355.
https://doi.org/10.1029/2019JG005355

Hines, M. E., Duddleston, K. N., Rooney-Varga, J. N., Fields, D., & Chanton, J. P.
(2008). Uncoupling of acetate degradation from methane formation in Alaskan
wetlands: Connections to vegetation distribution. *Global Biogeochemical Cycles*,
*22*(2). https://doi.org/10.1029/2006GB002903

Tøsdal, A., Urich, T., Frenzel, P., & Marianne, M. (2015). Metabolic and trophic
interactions modulate methane production by Arctic peat microbiota in response to
warming. *Proceedings of the National Academy of Sciences*, E2507–E2516.
https://doi.org/10.1073/pnas.1420797112

Updegraff, K., Bridgham, S. D., Pastor, J., & Weishampel, P. (1998). Hysteresis in the
temperature response of carbon dioxide and methane production in peat soils.
*Biogeochemistry*, *43*(3), 253–272. https://doi.org/10.1023/A:1006097808262

Yvon-Durocher, G., Allen, A. P., Bastviken, D., Conrad, R., Gudasz, C., St-Pierre, A., et
al. (2014). Methane fluxes show consistent temperature dependence across
microbial to ecosystem scales. *Nature*, *507*(7493), 488–491.
https://doi.org/10.1038/nature13164

Zhang, L., Liu, X., Duddleston, K., & Hines, M. E. (2020). The Effects of pH,
Temperature, and Humic-Like Substances on Anaerobic Carbon Degradation and
Methanogenesis in Ombrotrophic and Minerotrophic Alaskan Peatlands. *Aquatic
Geochemistry*, (0123456789). https://doi.org/10.1007/s10498-020-09372-0